# R-Ras-Akt axis induces endothelial lumenogenesis and regulates the patency of regenerating vasculature

Fangfei Li[1], Junko Sawada[1] & Masanobu Komatsu [1]

The formation of endothelial lumen is fundamental to angiogenesis and essential to the oxygenation of hypoxic tissues. The molecular mechanism underlying this important process remains obscure. Here, we show that Akt activation by a Ras homolog, R-Ras, stabilizes the microtubule cytoskeleton in endothelial cells leading to endothelial lumenogenesis. The activation of Akt by the potent angiogenic factor VEGF-A does not strongly stabilize microtubules or sufficiently promote lumen formation, hence demonstrating a distinct role for the R-Ras-Akt axis. We show in mice that this pathway is important for the lumenization of new capillaries and microvessels developing in ischemic muscles to allow sufficient tissue reperfusion after ischemic injury. Our work identifies a role for Akt in lumenogenesis and the significance of the R-Ras-Akt signaling for the patency of regenerating blood vessels.

[1] Tumor Microenvironment and Cancer Immunology Program, Cancer Center, Cardiovascular Metabolism Program, Center for Metabolic Origins of Disease, Sanford Burnham Prebys Medical Discovery Institute at Lake Nona, 6400 Sanger Road, Orlando, FL 32827, USA. Correspondence and requests for materials should be addressed to M.K. (email: mkomatsu@sbpdiscovery.org)

Endothelial cells (ECs) undergo tubular morphogenesis to create lumen and subsequently mature into functional blood vessels in order to generate new supply lines of oxygenated blood to hypoxic tissues. Blood vessel morphogenesis is often compromised in pathological or therapeutically induced angiogenesis resulting in the formation of defective neovasculature[1,2]. Despite its biological and potential clinical significance, the molecular mechanism of endothelial morphogenesis remains incompletely understood. In epithelial tubulogenesis, the microtubule cytoskeleton is crucial for defining cell polarity and supports the apical surface to establish the lumen structure[3]. The vesicle trafficking along microtubules allows transport of luminal proteins and membrane components to the apical plasma membrane thereby promoting cell polarization and lumen formation[3]. A recent in vitro study demonstrated the importance of the microtubule network for EC polarization, lumenization, and stabilization of the endothelial lumen structure[4]. During cell migration or division, the length of microtubules is dynamically altered by switching between polymerization and rapid depolymerization of tubulins, a critical property of microtubules known as dynamic instability[5]. On the other hand, the enduring stability of the microtubule architecture is thought to be required for the formation and maintenance of endothelial lumen[4,6]. The intracellular signaling pathways to confer microtubule stability to the endothelium of newly forming blood vessels have not been determined.

Akt is an important mediator of the angiogenic activity of vascular endothelial growth factor (VEGF). The phosphoinositide 3-kinase (PI3K)-Akt signaling mediates VEGF-induced nitric oxide production and angiogenic stimulation of ECs[7]. Akt can also disrupt endothelial barrier function and increase vascular permeability via an mTOR-dependent mechanism[8]. Ackah et al.[9] reported that the genetic ablation of *Akt1* in mice impaired reparative angiogenesis in the ischemic hindlimbs[9]. Thus, the $Akt1^{-/-}$ mice exhibited nearly obliterated limb reperfusion with little blood flow recovery[9]. However, the vessel density of the ischemic muscles of the $Akt1^{-/-}$ mice was found comparable to that of the control normoxic muscles of wild-type mice. This finding suggests that the Akt1 deficiency not only compromised EC's ability to proliferate and sprout, but also significantly impaired the morphogenesis of the new vessels, making these vessels inadequate for supporting the reperfusion and reoxygenation of the ischemic tissues.

The Ras family small GTPase protein, R-Ras, is a known activator of PI3K-Akt signaling[10]. It has been shown that R-Ras plays a critical role in the maturation of blood vessels in the regenerating vasculature[11]. R-Ras promotes pericyte association with nascent blood vessels and enhances the integrity of endothelial barrier while inhibiting excessive sprouting and branching of angiogenic vessels[11,12]. R-Ras is abundantly expressed in normal post-natal endothelium[12]. Although evolutionarily closely related[13,14], R-Ras is functionally distinct from the prototypic Ras proteins such as H-Ras and K-Ras. For instance, H-Ras mediates angiogenic response to VEGF via PI3K-Akt and Raf-Erk pathways to promote EC proliferation and migration[15–17]. In contrast, R-Ras activates the PI3K-Akt but not Raf-Erk signaling[10], and this PI3K-Akt activation does not seem to participate in the canonical proangiogenic Akt signaling since the gain-of-function of R-Ras limits vessel sprouting, branching, EC migration, and permeability[11,12,18]. This suggests that the Akt signaling elicited by R-Ras may have a distinct role in angiogenesis. Although the importance of Akt in angiogenesis has long been recognized[8,9,19–21], it is unclear whether Akt plays a role in endothelial morphogenesis. The deficiency of R-Ras in mice accentuates vascular malformation and disrupts blood perfusion of pathologically developing blood vessels in tumors[11]. On the other hand, R-Ras upregulation in ECs improves the perfusion of VEGF-induced microvessels[11]. The improved perfusion of these vessels may be due to the facilitation of proper endothelial morphogenesis.

Prompted by this idea, we investigated the potential contribution of the R-Ras-Akt axis to the molecular mechanism underlying endothelial lumenogenesis. We show that, when activated by R-Ras, Akt strongly and persistently stabilizes the microtubule cytoskeleton in ECs to promote endothelial lumen formation. We further demonstrate that this pathway is necessary for the production of functional patent blood vessels to enable sufficient reperfusion of ischemic muscles. These findings demonstrate the significance of R-Ras-Akt axis for ischemic tissue recovery and establish a role for Akt in endothelial lumenogenesis.

## Results

**R-Ras induces lumenization of endothelial sprouts**. We first analyzed the effect of R-Ras on endothelial morphogenesis in vitro in a fibrin gel three-dimensional (3-D) culture of ECs, which recapitulates the sequential steps of angiogenesis, i.e., sprouting, branching, and lumen formation (Fig. 1a, b). ECs were coated onto microbeads and embedded in fibrin gel with pericytes seeded on the top of the gel as feeder cells. The endothelial sprouts developed from the beads were analyzed at days 5–7. In this analysis, the mock-transduced control ECs developed a number of branching sprouts from the beads (Fig. 1a). Many of these sprouts showed that they have begun forming lumens (Fig. 1b, Supplementary Fig. 1, and Supplementary movie 1). However, these lumens are disconnected, and the sprouts are incompletely lumenized (Fig. 1b). The expression of constitutively active R-Ras (R-Ras38V) dramatically enhanced the EC lumenogenesis to form well-defined, uninterrupted lumen encased by the thin endothelial wall with flattened nuclei, a characteristic of intact capillary vessels and microvessels (Fig. 1a–c, Supplementary Fig. 1, and Supplementary movie 2). Some R-Ras38V-expressing EC sprouts were found anastomosed to form continuous luminal space that connects the lumens of two neighboring sprouts. While promoting lumen formation, the constitutively active R-Ras significantly reduced the number of sprouting and branching (Fig. 1a and Supplementary Fig. 2). Conversely, the silencing of endogenous R-Ras by short hairpin RNA severely disrupted EC lumenogenesis (Fig. 1a, b). Many of these ECs remained flat in the 3-D environment without undergoing morphogenesis into tubular structures. These results demonstrate a strong activity of R-Ras to induce lumen formation in endothelial sprouts.

**R-Ras stabilizes microtubules to promote lumenogenesis**. The microtubule cytoskeleton is crucial for establishing the epithelial cell polarity and lumenization of glandular epithelium[3]. A recent report suggests the importance of microtubules also for endothelial lumenogenesis[4]. We found that R-Ras is a key regulator of microtubule stability in ECs. This effect of R-Ras was demonstrated by markedly increased acetylation and detyrosination of α-tubulin (Fig. 1d, f, Supplementary Fig. 3a–c) as well as

**Fig. 1** R-Ras is required for microtubule stabilization and EC lumenogenesis in vitro. **a** Endothelial sprouts in 3-D fibrin gel culture were analyzed at day 5. Arrow, anastomosed adjacent sprouts forming a continuous lumen. **b** Mock-transduced control EC culture in higher magnification showed incomplete lumen formation in the sprouts (arrows). Lumen formation (arrowhead) was significantly accelerated and enhanced in R-Ras38V-transduced ECs. R-Ras knockdown by shRNA (shR-Ras) blocked tubular morphogenesis and lumen formation. **c** Size of individual lumen structure. The hallow structures of >5 μm length were considered as developing lumens. The area size of the individual lumen structure was determined by morphometry analysis and plotted on the graph. Sprouts grown from 10 EC-coated beads in two culture wells were analyzed for each group. Small vacuoles of <5 μm in length were not included in the analysis. Spouts without lumen were not examined. No lumen was formed by R-Ras-silenced ECs. $p < 1 \times 10^{-5}$. **d–i** R-Ras stabilizes endothelial microtubules. Immunofluorescence of the total (green) and acetylated α-tubulin (**d**) or delta 2-tubulin (**e**) (magenta). **f** The ratio of post-translationally modified α-tubulin (acetylated α-tubulin or delta 2-tubulin) to the total α-tubulin was quantified from immunofluorescence staining of the cells. **p < 0.001. **g** Immunofluorescence of α-tubulin and microtubule end-binding protein (EB1) to indicate the (+) ends of microtubules. Thin gray lines indicate the outlines of the cell membrane. Lower magnification images available in Supplementary Fig. 4. **h** In-gel staining of endothelial sprouts for total (green) and acetylated (red) α-tubulin in 3-D culture (day 7) is shown by confocal sectional images. Yellow color indicates co-staining. **i** Higher magnification to show the pattern of extending microtubules. Acetylated microtubule fibers are depicted by yellow lines in the schematic representation of an R-Ras38V-expressing sprout in cross-section. **j** Nocodazole was added at 10 μM to 5-day-old culture of mock or R-Ras38V-transduced EC sprouts. Images of the sprouts were taken before (Noc−) and 1.5 h after (Noc+) nocodazole treatment. Arrows indicate two discontinuous lumens in a control sprout and a complete uninterrupted lumen in an R-Ras38V$^+$ sprout. Scale bars, 75 μm (**a**), 50 μm (**b, h, j**), 20 μm (**d, e**)

pronounced extension of microtubules that reach the cell membrane (Fig. 1g, Supplementary Fig. 4). Moreover, there was a considerable accumulation of delta 2-tubulin, a modified form of α-tubulin, which indicates the formation of long-lasting, highly stable microtubule cytoskeleton (Fig. 1e, f)[22]. A biochemical cell fractionation analysis demonstrated an accumulation of α-tubulin

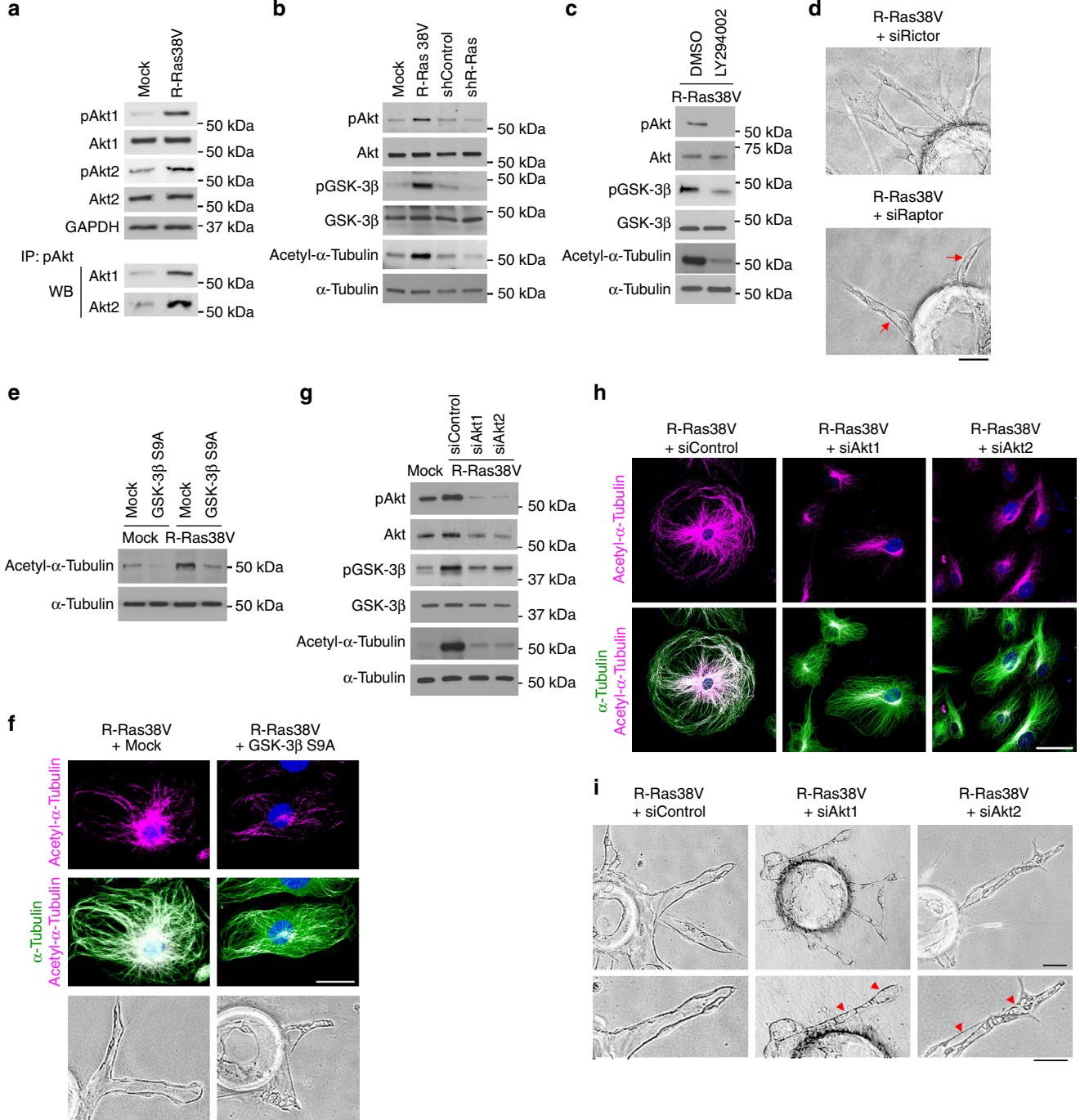

**Fig. 2** Akt mediates R-Ras-dependent microtubule stabilization and lumenogenesis. **a** R-Ras activates Akt isoforms. Ser473 phosphorylation of Akt1 and Akt2 were determined by western blot using phospho-specific antibody for each isoform as well as by immunoprecipitation of Ser473-phosphorylated Akt followed by western blot for each Akt isoform. **b** The effects of R-Ras on Akt Ser473 and GSK-3β Ser9 phosphorylation and α-tubulin acetylation were examined by western blot. **c** PI3K inhibition by LY294002 blocks the effects of R-Ras38V. **d** Endothelial sprouts in 3-D cultures of R-Ras38V-transduced, Rictor, or Raptor-silenced ECs. ECs were transduced with R-Ras38V followed by Rictor or Raptor knockdown using siRNA (siRictor or siRaptor) before embedded in fibrin gel. Arrows, R-Ras-induced lumenogenesis was unaffected by Raptor knockdown. **e, f** Constitutive activation of GSK-3β blocks R-Ras-dependent microtubule stabilization and lumenogenesis. ECs were first transduced with/without R-Ras38V and subsequently transduced with/without GSK-3β S9A. Acetylation of α-tubulin was analyzed by western blot (**e**) and immunofluorescence (**f**) and lumenogenesis examined in 3-D culture (**f**). **g–i** Akt1 and Akt2 are essential to the R-Ras-mediated GSK-3β inhibition, microtubule stabilization, and lumenogenesis in vitro. Either Akt isoform was silenced in R-Ras38V-transduced ECs by siRNA (siAkt1 or siAkt2). GSK-3β phosphorylation and α-tubulin acetylation were examined by western blot (**g**) or western blot and immunofluorescence (**g, h**). The effect of Akt silencing on R-Ras-mediated EC lumenogenesis was examined in 3-D cultures (**i**). Arrowheads, severely disrupted lumen formation. Scale bars, 50 μm (**d**, **f** 3-D culture, **h**, **i**), 25 μm (**f** 2-D culture)

in the detergent-insoluble fraction similarly to the effect of a microtubule-stabilizing agent, paclitaxel, further supporting the role of R-Ras in the formation of stable microtubule cytoskeleton in ECs (Supplementary Fig. 3d, e).

The effect of R-Ras on microtubules was also observed in endothelial sprouts in the 3-D culture that exhibited enhanced lumenogenesis and uninterrupted capillary-like tubular structures (Fig. 1h). In the lumenized sprouts, microtubules extended from

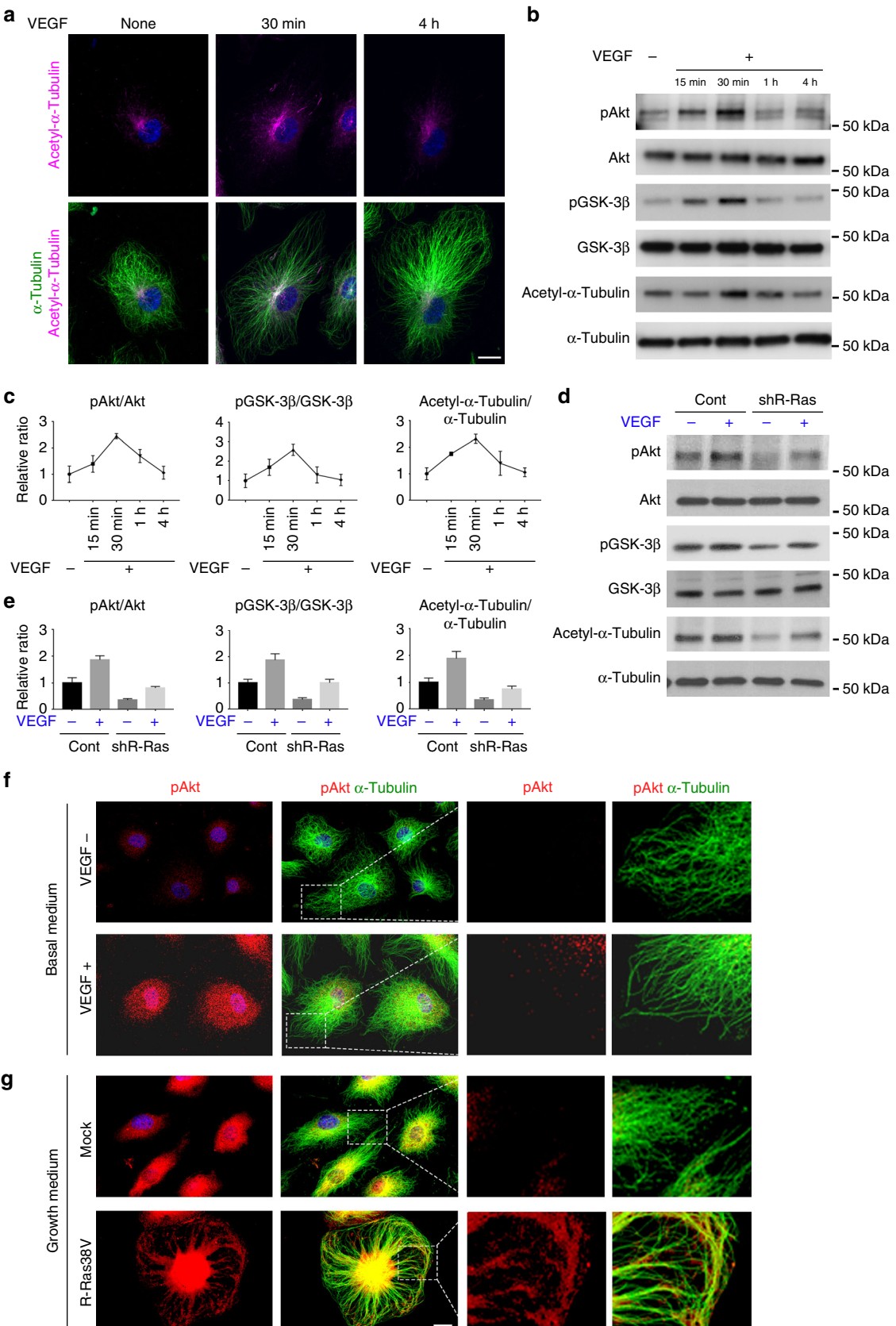

the microtubule organizing center positioned at the apical side of the nucleus, indicating the polarization of ECs (Fig. 1i, Supplementary Fig. 5). On the other hand, R-Ras knockdown destabilized and shorten the microtubules and severely impaired lumenogenesis demonstrating the importance of endogenous R-Ras (Fig. 1d–h). The ability of R-Ras to promote endothelial lumenogenesis is microtubule-dependent as microtubule disruption by nocodazole collapses preformed lumens (Fig. 1j).

**Akt mediates the effect of R-Ras**. We next investigated the mechanism of microtubule stabilization by R-Ras. Glycogen synthase kinase (GSK)-3β destabilizes microtubules and disrupts polarization of neurons[23–25]. Since GSK-3β is a substrate for Akt[26], the effect of R-Ras may be mediated by activation of a PI3K-Akt pathway, which phosphorylates and inhibits GSK-3β, leading to microtubule stabilization in ECs. Supporting this hypothesis, R-Ras38V significantly increased phosphorylation of Akt1 and Akt2 isoforms and GSK-3β concomitantly with acetylation of α-tubulin (Fig. 2a, b, Supplementary Fig. 6a) indicating the activation of Akt isoforms, inhibition of GSK-3β, and stabilization of microtubules. R-Ras knockdown had opposite effects demonstrating the importance of endogenous R-Ras for this pathway (Fig. 2b, Supplementary Fig. 6a). The inhibition of PI3K by LY294002 nullified the effect of R-Ras on Akt activation, GSK-3β phosphorylation, and microtubule stabilization (Fig. 2c, Supplementary Fig. 6b). Furthermore, the disruption of mTOR complex 2 (mTORC2) upstream of Akt by Rictor knockdown blocked the R-Ras38V-induced EC lumenogenesis (Fig. 2d).

To examine downstream of Akt, we disrupted mTORC1 by Raptor knockdown. However, this did not block R-Ras38V from inducing lumenogenesis (Fig. 2d) suggesting that the effect of R-Ras is independent of the Akt-mTORC1 signaling. In contrast, the expression of a constitutively active GSK-3β mutant (GSK-3β S9A), which is resistant to phosphorylation and inhibition by Akt, blocked the R-Ras-dependent microtubule stabilization and endothelial lumenogenesis while severely damaging the formation of sprouts (Fig. 2e, f, Supplementary Fig. 6c). These results suggest that R-Ras-Akt axis stabilizes microtubules and promotes lumenogenesis of EC sprouts at least in part via Akt-dependent GSK-3β inhibition but not via mTORC1 activation.

Akt1 and Akt2 are the major isoforms of Akt expressed in ECs[9]. The relative contribution of Akt isoforms for this pathway was investigated by individually silencing Akt1 or Akt2. The RNAi of either isoform nullified the effect of R-Ras38V to induce GSK-3β phosphorylation and stabilize microtubules (Fig. 2g, h, Supplementary Fig. 6d). Consistently, the silencing of either Akt isoform disrupted the EC lumenogenesis promoted by R-Ras38V (Fig. 2i). These results demonstrate that both Akt isoforms contribute critically to the lumenogenesis-promoting effect of R-Ras. The silencing of either Akt isoform also disrupted lumenogenesis in wild-type ECs expressing endogenous R-Ras (Supplementary Fig. 7).

**Differential effects of VEGF and R-Ras signaling on Akt**. VEGF-A activates Akt through VEGF receptor-2 and elicits a potent angiogenic stimulation in ECs[15]. However, the excess VEGF stimulation can disrupt EC lumenogenesis[2,27]. We stimulated ECs with VEGF-A and analyzed its effect on Akt, GSK-3β, and microtubules. The VEGF-dependent Akt activation was not associated with strong and persistent microtubule stabilization albeit apparent transient increase of microtubule polymerization (Fig. 3a). The ECs stimulated with VEGF-A showed a transient activation of Akt. This activation was observed only during the first 15–30 min after the cell stimulation with VEGF-A (Fig. 3b, c). The Ser 9 phosphorylation of GSK-3β that indicates GSK-3β inhibition was also observed only during this period coinciding the Akt activation (Fig. 3b, c). VEGF-A stimulation of R-Ras-silenced ECs showed that these transient effects of VEGF-A is independent of R-Ras (Fig. 3d, e), supporting the idea that the VEGF-Akt and R-Ras-Akt axes are separate pathways. We also examined the effect of angiopoietin-1 (Ang-1) as Ang-1 is important for remodeling and maturation of nascent vessels[28]. Ang-1 induced a sharp but transient increase of Akt and GSK-3β phosphorylation and concomitant α-tubulin acetylation, which peaked at 15 min (Supplementary Fig. 8a, b). In contrast, Ang-1 moderately increased R-Ras activity. (Supplementary Fig. 8c). However, the temporal pattern of R-Ras activation does not seem to match the pattern of Ang-1-dependent Akt/GSK-3β phosphorylation that declined sharply after 15 min. Notably, the addition of Ang-1 did not enhance endothelial lumenogenesis in 3-D culture (Supplementary Fig. 8d, e). These results suggest that Ang-1 may be insignificant for the R-Ras-Akt axis and endothelial lumenogenesis. The study with R-Ras silencing demonstrated a significant contribution of endogenous R-Ras to the basal levels of Akt activity, GSK-3β phosphorylation, and α-tubulin acetylation without growth factor stimulation, suggesting a relatively high basal activity of endogenous R-Ras in ECs (Fig. 3d, e).

We next analyzed the subcellular localization of Akt upon activation by VEGF or R-Ras. The immunofluorescence staining of ECs indicated that the Ser 473-phosphorylated Akt localizes at the perinuclear region upon cell stimulation with VEGF-A (Fig. 3f). This localization pattern of activated Akt has been demonstrated previously[29,30], and it is known to be required for fully mediating the effect of the growth factor signaling[31]. In contrast, the expression of R-Ras38V produced a distinct pattern of Akt localization in ECs. In these cells, the accumulation of activated Akt was found closely associating with the microtubule cytoskeleton in addition to the accumulation in the perinuclear region (Fig. 3g). The association of activated Akt with microtubules was observed along the microtubule fibers, all the way to the (+) end of microtubules positioned at the membrane periphery. Such an association was not found in the ECs stimulated with VEGF (Fig. 3f). Corroborating these results, R-Ras38V increased accumulation of activated Akt in the detergent-insoluble cytoskeletal fraction (Supplementary Fig. 9). The silencing of R-Ras reduced activated Akt from this fraction, suggesting the importance of endogenous R-Ras signaling for the cytoskeletal association of activated Akt (Supplementary Fig. 9). In comparison, activated Akt in the cytoskeletal fraction remained at a basal level upon stimulation by VEGF. In total, these results suggest that the differential effects of Akt on EC

**Fig. 3** VEGF and R-Ras signaling exert differential effects on Akt and microtubule. ECs were cultured in low-serum basal media (2% horse serum without growth factor supplements) for overnight and stimulated with (+) or without (−) 50 ng/ml VEGF-A for indicated time. Akt (Ser473) and GSK-3β (Ser9) phosphorylation and α-tubulin acetylation were analyzed by immunofluorescence (**a**) and/or western blot (**b**). **c** Levels of phosphorylated Akt, GSK-3β, and acetylated α-tubulin were quantitated by densitometry and normalized to the corresponding total protein levels. **d** Control or R-Ras-silenced ECs were stimulated with VEGF-A, and Akt/GSK-3β phosphorylation and α-tubulin acetylation were analyzed. **e** Western blots of phosphorylated Akt, GSK-3β, and acetylated α-tubulin were normalized to corresponding total protein levels. **f**, **g** Immunofluorescence of phospho-Akt (red) and total α-tubulin (green). ECs were cultured in basal media and stimulated with (+) or without (−) 50 ng/ml VEGF-A for 30 min (**f**). Mock or R-Ras38V-transduced ECs (**g**). Higher magnification of the boxed area is also shown. The graphs show the combined results of at least three independent experiments. Scale bars, 20 μm

activities and lumenogenesis may be attributed to the duration of the Akt signaling and/or distinct subcellular location depending on the activation pathways.

**R-Ras is crucial to vessel lumenization during reparative angiogenesis.** Next, we investigated the role of the R-Ras-Akt axis in endothelial lumenogenesis in vivo. This pathway may be important primarily for post-natal regenerative angiogenesis since the expression of R-Ras is not detected during embryonic vascular development[12]. We, therefore, investigated the R-Ras-Akt pathway in the context of reparative angiogenesis in mouse ischemic hindlimbs. ECs of intramuscular capillaries and microvessels express R-Ras while skeletal myocytes show no detectable R-Ras in wild-type mice before and after ischemia induction by femoral

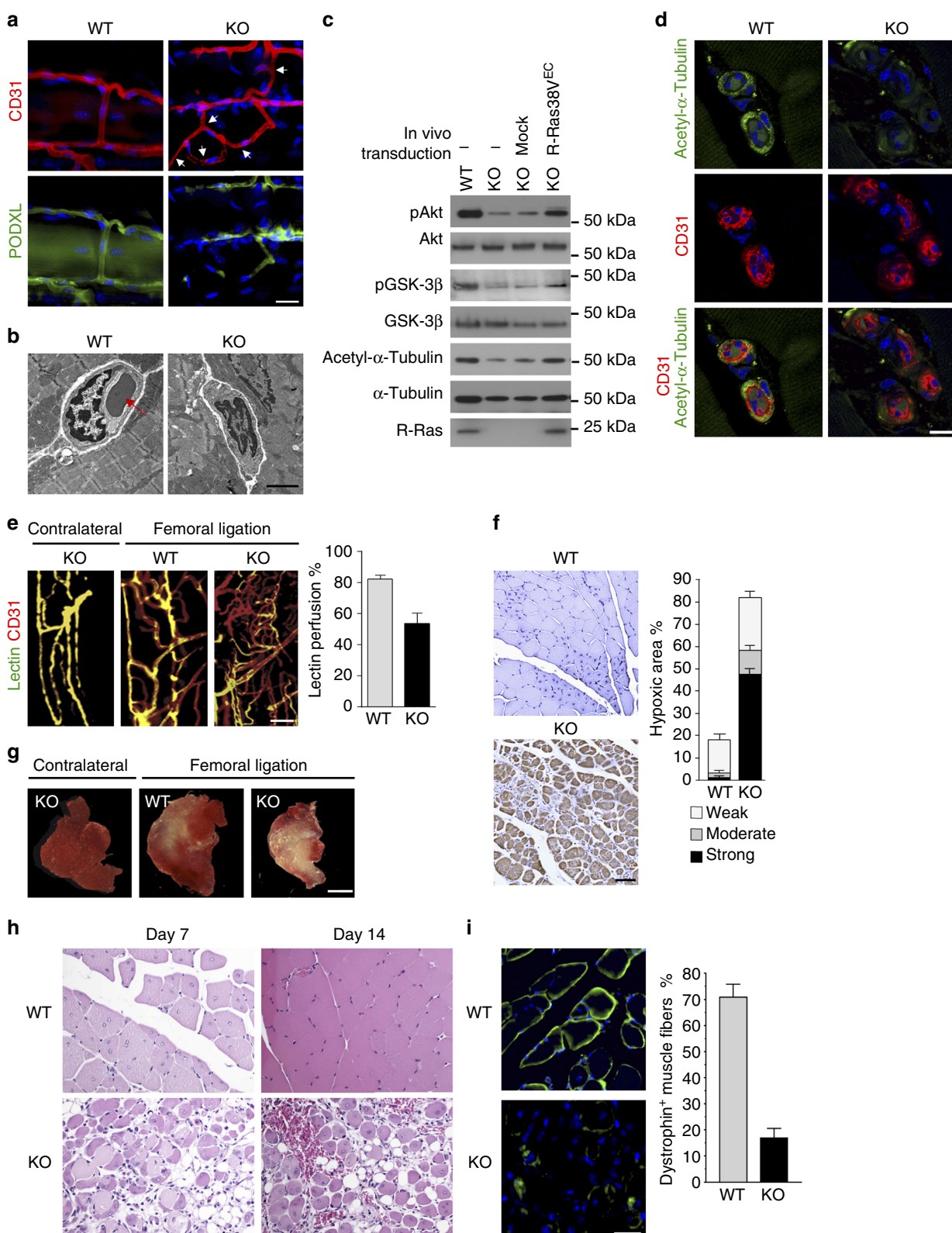

artery ligation (Supplementary Fig. 10). Upon ischemia induction, $Rras^{-/-}$ (R-Ras knockout, KO) mice exhibited more robust angiogenesis than wild-type mice, producing highly chaotic neovasculature with sustained EC proliferation in the gastrocnemius (GC) muscle (Supplementary Fig. 11) as expected from the absence of R-Ras to attenuate excessive angiogenesis[11]. However, the pattern of immunostaining for podocalyxin (PODXL), which highlights endothelial lumen[1,27,32–34], was frequently disrupted in these vessels indicating a deficiency in lumenogenesis (Fig. 4a). Ultrastructural analyses by electron microscopy confirmed that numerous R-Ras KO vessels developed in response to ischemia indeed lacked lumens (Fig. 4b). The Akt and GSK-3β phosphorylation and α-tubulin acetylation were all considerably reduced in the ECs of these vessels (Fig. 4c, d) substantiating the effect of R-Ras silencing observed in the cultured ECs and EC sprouts (Figs. 1 and 2).

The defective lumenogenesis of R-Ras KO vessels was associated with significantly reduced blood perfusion of the neovasculature (Fig. 4e), impaired blood flow recovery in the ischemic foot (Supplementary Fig. 12), and substantial muscle hypoxia (Fig. 4f) despite increased EC proliferation and vascularity (Supplementary Fig. 11). The strong and sustained hypoxia resulted in the failure of muscle recovery, extensive tissue necrosis, and severe muscle mass loss (Fig. 4g, h, i, Supplementary Fig. 13) highlighting the importance of R-Ras-dependent endothelial lumenogenesis for the reperfusion of ischemic muscles.

We next conducted an EC-targeted *RRAS* gene therapy to rescue lumenogenesis in R-Ras KO vessels. For this purpose, we generated a lentivirus vector, which carries R-Ras38V under the mouse vascular endothelial (VE)-cadherin (*Cdh5*) promoter[35] for EC-specific expression of R-Ras38V (R-Ras38V^EC). The virus was injected into the GC muscles of R-Ras KO mice at 3 days after the ischemia induction to allow initial angiogenic sprouting to take place without attenuation by R-Ras (Fig. 5a, Supplementary Fig. 14). This treatment locally restored the endothelial lumen formation (Fig. 5b, c) while suppressing chaotic vessel production (Fig. 5d). The restoration of EC lumenogenesis resulted in improved vessel perfusion and muscle recovery after ischemia (Fig. 5e, f, g). These findings establish a central role of R-Ras in vessel lumenization to generate patent blood vessels in ischemic tissues, which is essential to the recovery from ischemic injuries.

**R-Ras-Akt signaling promotes vessel lumenogenesis**. We found that the lumen restoration and perfusion recovery by the EC-targeted *RRAS* gene therapy were associated with elevated Akt and GSK-3β phosphorylation and microtubule stabilization in the ECs (Figs. 4c and 6b). In comparison, EC-targeted in vivo transduction of R-Ras38V D64A mutant (Fig. 6a), which is

constitutively active but incapable of activating Akt due to a mutation in the PI3K-binding site[36], did not stabilize microtubules, or rescue lumenogenesis of R-Ras KO vessels, supporting the importance of Akt as a downstream mediator of R-Ras signaling (Fig. 6b–d, Supplementary Fig. 15). To further demonstrate the requirement of Akt activity for the R-Ras effect, we treated the mice with an Akt inhibitor MK-2206 at a dose used for cancer treatments in preclinical studies[37], from 1 day after the R-Ras38V lentivirus injection (Fig. 6a). In this study, the Akt inhibition blocked R-Ras38V from rescuing EC lumenogenesis and vessel perfusion in the R-Ras KO mice (Fig. 6c, d). Consistent with this finding, we observed that R-Ras38V failed to stabilize microtubules in the ECs when mice were treated with the Akt inhibitor (Fig. 6b). Taken together, these results demonstrate for the first time an Akt-dependent mechanism of lumenogenesis.

Akt activation is downstream of the angiogenic stimulation by VEGF-A[15]. Using wild-type mice, we investigated how a treatment of ischemic limbs with VEGF-A adenovirus (Ad-VEGF) gene therapy affects EC lumenogenesis while promoting EC sprouting. The injection of Ad-VEGF increased the vascularity of the ischemic GC muscles as expected (Supplementary Fig. 16). However, the PODXL^+ lumen formation was found diminished substantially in these new vessels (Supplementary Fig. 16), indicating that VEGF-A stimulation alone does not sufficiently promote endothelial lumenogenesis. Interestingly, the insufficient lumenogenesis of the VEGF-induced vessels was associated with downregulation of R-Ras in the ECs (Supplementary Fig. 16). In the analysis of cultured ECs, the cell stimulation with VEGF-A transiently activated Akt but did not promote strong or sustained microtubule stabilization or GSK-3β inhibition (Fig. 3), which was in sharp contrast to the effect of the R-Ras-dependent Akt activation (Fig. 2). These results demonstrate differential roles of the R-Ras and VEGF axes of Akt signaling in angiogenesis (Fig. 7).

## Discussion

Although endothelial lumenogenesis has long been investigated, the molecular and cellular mechanism of this important biological process has not been clearly understood. In this study, we identified a role for Akt in lumenogenesis. We found that Akt activation by R-Ras is a critical signaling event for the lumen formation during reparative angiogenesis. In the absence of R-Ras, many lumen-less vessels are generated in the ischemic muscles. These defective vessels cannot participate in the blood circulation system and make no contribution to supplying oxygen and nutrients to the ischemic lesions for the recovery. Consequently, the R-Ras-deficient mice suffered extensive tissue necrosis and substantial muscle mass loss. The EC-targeted restoration of R-Ras signaling, which rescued the vessel

**Fig. 4** Defective EC lumenogenesis and impaired muscle reperfusion in R-Ras deficiency. Hindlimb ischemia was induced in wild-type (WT) and R-Ras KO (KO) mice by left femoral artery ligation, and GC muscles were analyzed 14 days later. **a** Immunofluorescence of CD31 and PODXL to identify lumenized vessels. Arrows indicate lumen-less vessels formed in the ischemic GC muscles of R-Ras KO mice. **b** Transmission electron microscopy confirmed the absence of lumen structures in numerous R-Ras KO vessels developed in response to ischemia. Muscle were sectioned perpendicular to the muscle fibers. Arrow, a circulating erythrocyte indicates normal lumen formation in a wild-type vessel. **c** Analyses of Akt (Ser473) and GSK-3β (Ser9) phosphorylation and α-tubulin acetylation in ECs isolated from ischemic GC muscles at day 14. ECs were also isolated from a separate set of R-Ras KO mice, which received lentivirus injection into the GC muscles for EC-specific expression of R-Ras38V (R-Ras38V^EC) via in vivo transduction. **d** α-Tubulin acetylation in the endothelium of intramuscular vessels. **e** Lectin perfusion (green) into intramuscular vessels (red) was determined in the whole-mounted GC muscle fascicles. Yellow color (green/red double-staining) indicates blood perfused vessels. Lectin perfusion % = lectin^+CD31^+ area/total CD31^+ area × 100, $p <$ 0.01, $n = 5$. **f** Analysis of hypoxia in GC muscles at day 7 by hypoxyprobe-1™ staining (brown). Thresholds were set empirically for identifying the area with strong, moderate, or weak staining and presented as % of total muscle area examined. $p = 8 \times 10^{-6}$, $n = 5$. **g** Muscle viability was assessed at day 14 by staining the slices of unfixed GC muscles with 2,3,5-triphenyltetrazolium chloride, which stains viable tissues in red. The infarct areas are unstained (pale yellow/white). **h** H&E staining of GC muscle sections. **i** Dystrophin immunostaining (green) of GC muscle cross-sections to quantify functional muscle fibers. The number of dystrophin^+ muscle fibers/total muscle fibers (%) was determined in non-necrotic area. $p < 10^{-4}$, $n = 5$. Scale bars, 25 μm (**a**, **h**), 2 μm (**b**), 10 μm (**d**), 50 μm (**f**), 2 mm (**g**)

lumenization and perfusion in these mice, confirmed the significance of the endothelial R-Ras-Akt pathway for the tissue recovery.

Our results indicate that the R-Ras-Akt signaling is functionally distinct from the VEGF-induced mTORC1 or nitric oxide-mediated canonical angiogenic Akt signaling that leads to vessel sprouting or vascular permeability[38,39]. We showed that the R-Ras-Akt signaling induces a strong and persistent microtubule stabilization in ECs to promote the lumen formation. Our data indicate that this effect is mediated at least in part via inhibition of GSK-3β by Akt but not via mTORC1 activation. Furthermore, we showed that Akt activity is indispensable for the lumenogenesis-inducing effect of R-Ras. The significant role of the microtubule cytoskeleton is well recognized for the epithelial

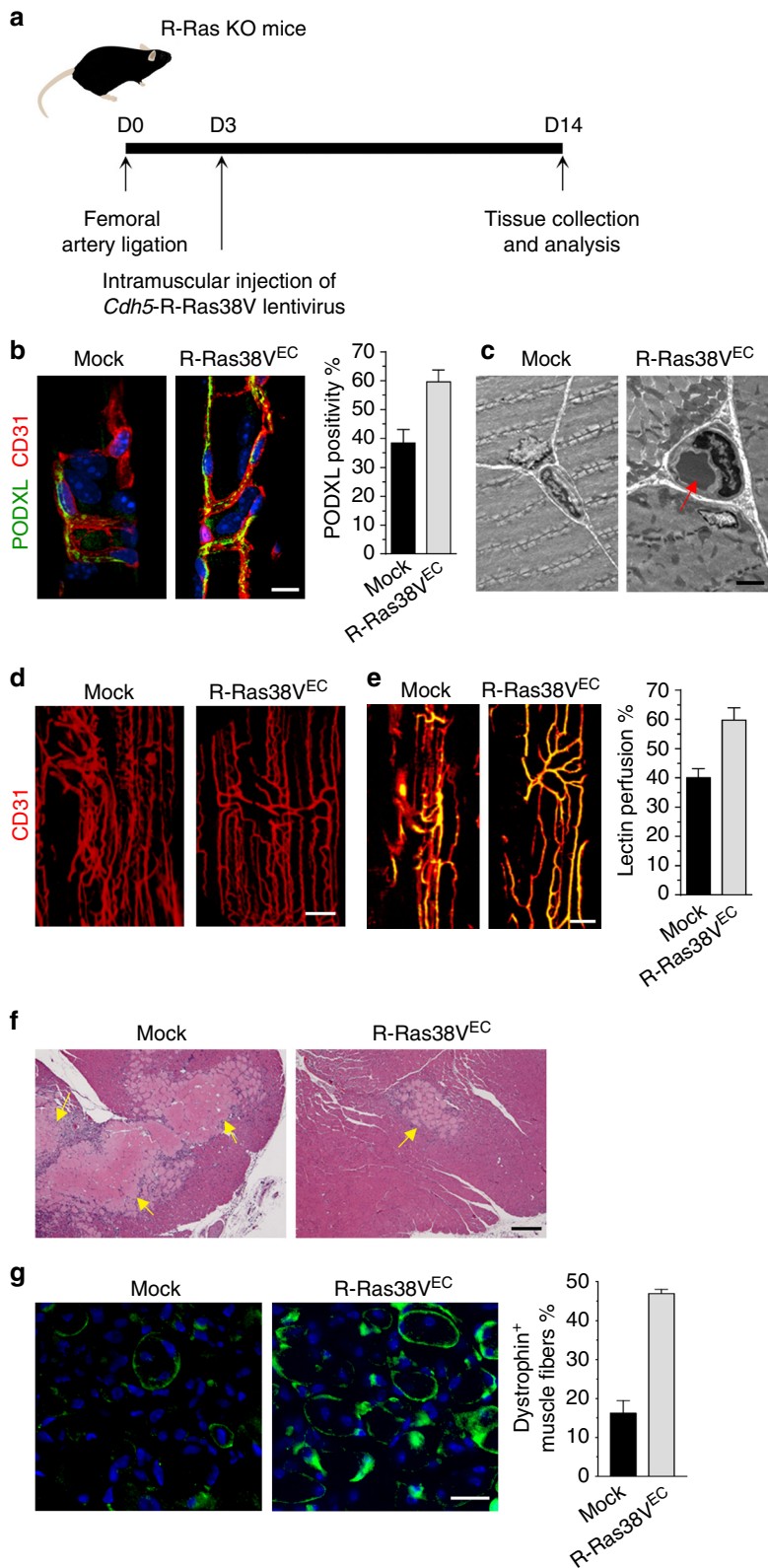

lumen formation and maintenance[3], and its importance has also been demonstrated in vitro for endothelial lumenogenesis[4,6]. There are several models for the potential cellular mechanism of lumen formation, including cell hallowing and cord hallowing mechanisms involving vacuole fusions[3,40]. In any case, intracellular vesicle transport along microtubules toward the apical surface is thought to be necessary for the cell polarization and lumen expansion during tubulogenesis[3]. It is likely that R-Ras-Akt signaling secures the vesicle transport and drive lumen formation through providing a stable microtubule network. The stabilized microtubules reaching all the way to the apical membrane would certainly be an advantage for securing the transportation of luminal membrane materials. In ischemic muscles, PODXL was undetectable in many R-Ras-deficient vessels by tissue immunofluorescence although the total PODXL protein level was unaltered by R-Ras in vitro (Supplementary Fig. 17). This is probably because the detectable fluorescence signal declines with diffused localization of PODXL associated with the lack of cell polarity in these vessels. This notion is consistent with the role of microtubules in transporting PODXL to the apical surface in polarized ECs. Unlike the basal and lateral sides of the cell, which are structurally supported by the basement membrane extracellular matrix and neighboring cells, the apical side of the cell has no extracellular structural support because the lumen is filled with fluid. The microtubule cytoskeleton is thought to provide, not only the vesicle-transporting cables for lumen creation, but also an essential architectural support for the maintenance of the lumen structure[4,6]. Our results indicate that the R-Ras-Akt axis is also important for this role of the microtubules. We showed that the disruption of microtubules by nocodazole immediately collapsed preformed lumens that have already been established by the expression of constitutively active R-Ras. These observations underscores the importance of the microtubule cytoskeleton in ECs as well as the R-Ras-Akt signaling to stabilize it for maintaining the luminal structure of patent blood vessels.

Although our work clearly demonstrated the importance of the microtubule regulation by R-Ras-Akt axis, these findings do not preclude potential contributions of other pathways of R-Ras to vessel lumenogenesis. Integrin adhesion to the extracellular matrix and VE-cadherin stabilization are important for epithelial and EC polarity and lumenogenesis[27,41,42]. R-Ras enhances cell-matrix and cell–cell adhesions via integrin activation[43] and VE-cadherin stabilization[11]. These and microtubule-stabilization effects of R-Ras presumably work in concert to develop functional blood vessels. Since R-Ras expression is undetected during embryonic vascular development and R-Ras KO mice develop without gross vascular abnormalities[12], the importance of this pathway may be unique to post-natal regenerative angiogenesis such as the revascularization of ischemic tissues, as we observed in this study, and tumor angiogenesis. Whether Akt activation by other mechanisms plays a role in microtubule stabilization or vessel morphogenesis during embryonic vascular development is unknown.

We showed that the R-Ras-Akt signaling stabilizes microtubules and promotes endothelial lumenogenesis, but VEGF-induced Akt signaling does not have these effects. In contrast, VEGF-dependent Akt activation induces angiogenic responses in ECs (i.e., proliferation, migration, and increased permeability), but R-Ras-Akt signaling does not. Thus, the VEGF-Akt signaling is largely skewed toward the induction of vessel sprouting and permeability, but it is insufficient for driving lumenogenesis or stabilizing nascent vessels by itself. The R-Ras-Akt signaling, on the other hand, promotes lumenogenesis and supports the lumen structure. R-Ras exerts these effects while limiting nonproductive angiogenesis and plasma leakage by enhancing vessel integrity[11,12]. We propose that the VEGF and R-Ras axes of Akt signaling are complementary to each other, and both are necessary to generate functional blood vessels for tissue recovery from ischemic injury (Fig. 7).

Our finding raises a new question as to how Akt signaling functionally diverges depending on the activation mechanism. While we do not have a precise answer, one possible explanation may be related to the subcellular localization of signaling molecules. H-Ras and R-Ras are known to localize in different membrane compartments upon activation[44]. It is conceivable that the differential subcellular localization of activated Akt confers distinct biological outcomes. This notion is supported by the observation that Akt accumulates closely associating with the microtubule cytoskeleton upon activation by R-Ras and that this pattern of Akt localization is not observed for the activation by VEGF or growth factor-rich media. Alternatively, Akt may form a complex with a different set of signaling effectors to produce distinct effects depending on the activation pathway. Another possibility is that the duration of Akt activation, for instance transient vs. chronic activity, may produce different effects. We showed that VEGF alone is insufficient to support microtubule stabilization or to promote endothelial lumenogenesis despite its ability to strongly activate Akt and inhibit GSK-3β for a short time. It is expected that such a short effect would not contribute to the stability of the microtubule network required for the lumen formation and maintenance since sustained GSK-3β inhibition is necessary for such a stability. Considering the known functions of R-Ras in vessel wall stability that likely require constant effects of R-Ras[11,12], it is conceivable that the basal level of endogenous R-Ras activity may be relatively high in ECs providing constant signaling. Indeed, there is a sizable contribution of endogenous R-Ras to the Akt activity (pAkt), GSK-3β phosphorylation, and α-tubulin acetylation in ECs without growth factor stimulation. These potential mechanisms of differential Akt effects are not mutually exclusive.

There have been intensive investigations on the VEGF-based therapeutic angiogenesis as a treatment strategy for the ischemic conditions such as coronary artery disease and peripheral arterial disease. More than 25 phase II and III clinical trials of therapeutic angiogenesis have been conducted to date; however, all of these trials have failed to offer significant benefit to the patients[45,46]. VEGF therapies can result in abnormal vascularization with disorganized and primitive vascular plexuses[2,46,47]. These vessels provide no significant improvements in ischemic tissue reperfusion[45,46,48,49]. Thus, clinical observations also indicate that VEGF

**Fig. 5** *RRAS* gene delivery to ECs rescues vessel lumenogenesis and muscle reperfusion. **a** In vivo transduction and treatment schedule. The lentivirus carrying pLenti6/*Cdh5*-R-Ras38V expression vector was injected into GC muscles 3 days after ligation. GC muscle were analyzed at day 14. **b** Immunofluorescence of CD31 and PODXL to identify lumenized vessels in GC muscles after the lentivirus injection for EC-specific expression of R-Ras38V (R-Ras38V[EC]). PODXL positivity % (PODXL[+]CD31[+] area/CD31[+] area × 100) was determined to assess the fraction of lumenized vessel area. **c** Transmission electron microscopy of the GC muscles confirmed the increase in vessel lumenization upon R-Ras38V[EC] transduction. Arrow, a circulating erythrocyte found in the vessel lumen indicating normal lumen formation. **d** CD31 staining of whole-mounted GC muscle fascicles. **e** Analysis of vessel perfusion in whole-mounted GC muscle fascicles. Yellow color indicates lectin perfused vessels. **f** H&E staining of GC muscle sections. Arrows, necrotic areas. **g** Dystrophin immunostaining (green) of GC muscle cross-sections to quantify functional muscle fibers. The number of dystrophin[+] muscle fibers/total muscle fibers (%) was determined in non-necrotic area. $p < 10^{-4}$, $n = 5$. Scale bars, 25 μm (**b**), 3 μm (**c**), 100 μm (**d**), 50 μm (**e**), 150 μm (**f**)

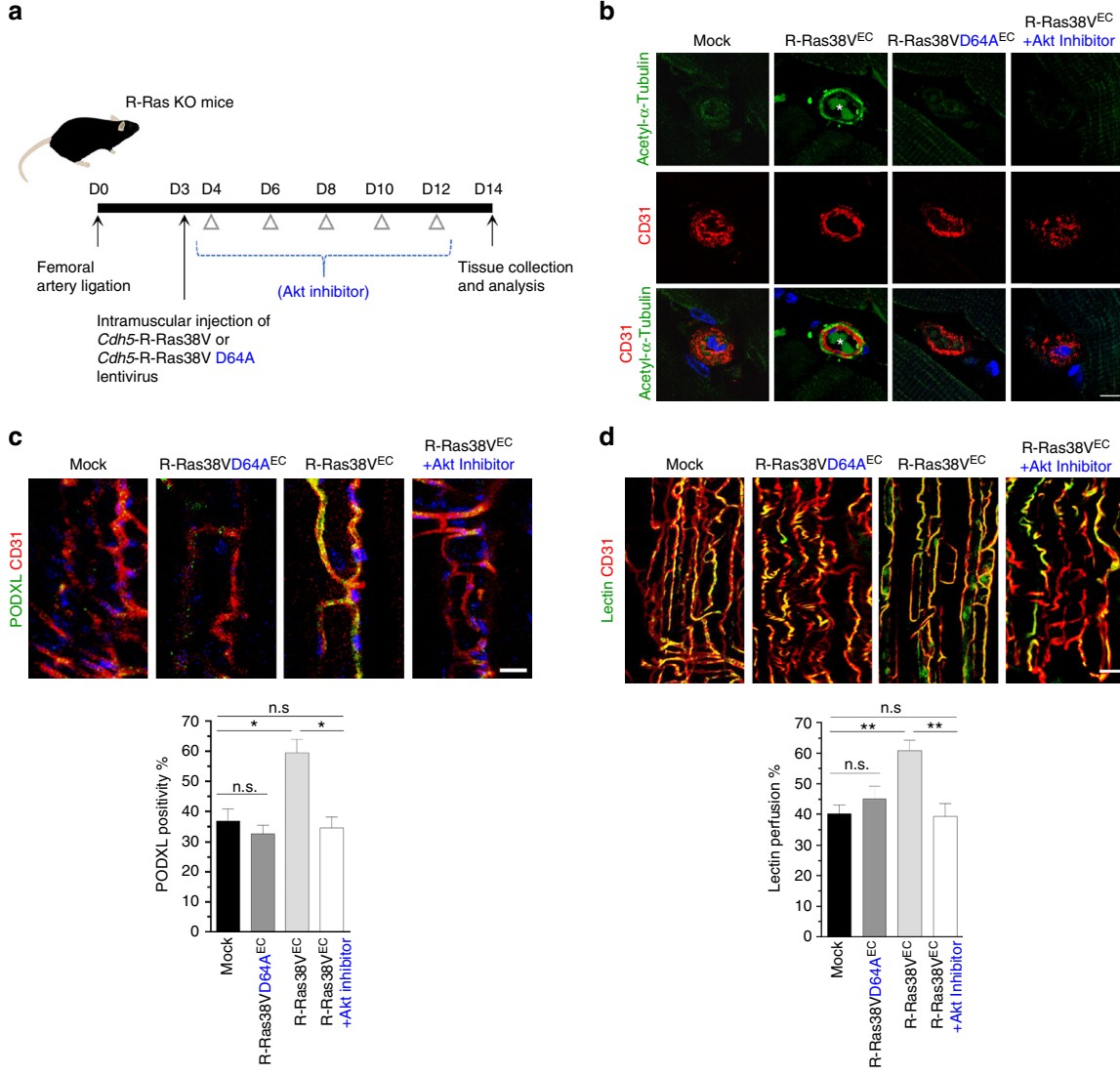

**Fig. 6** R-Ras promotes formation of lumenized functional blood vessels via Akt. **a** In vivo transduction and treatment schedule. One day after lentivirus injection into GC muscles, R-Ras KO mice were treated with or without Akt inhibitor MK2206 by gavage every 2 days. GC muscle were analyzed at day 14. **b** Immunofluorescence of GC muscle cross-sections to examine α-tubulin acetylation in the endothelium of intramuscular vessels. *Blood cells in the lumen. **c** CD31 and PODXL immunofluorescence of the GC muscles that received lentivirus injection for EC-specific R-Ras38V D64A or R-Ras38V expression and subsequently treated with Akt inhibitor MK2206. *$p < 0.01$, n.s., not significant. **d** Analysis of vessel perfusion in whole-mounted GC muscle fascicles. Yellow color indicates lectin perfused vessels. **$p < 0.02$. Scale bars, 10 μm (**b**), 25 μm (**c**), 50 μm (**d**)

alone is insufficient to promote the regeneration of fully functional neovasculature. We showed in this study that, although VEGF gene therapy potentiates the angiogenic response in the ischemic muscles, it is inefficient in promoting the formation of patent blood vessels with lumenized endothelium. Moreover, this inefficiency is associated with R-Ras downregulation in the ECs. These findings suggest that simultaneous or sequential enhancement of VEGF and R-Ras pathways could be used as a strategy to generate increased number of patent blood vessels to help reestablish capillary/microcirculation in the ischemic tissues. The new role of the R-Ras-Akt axis we identified in this study may be exploited for effective tissue reperfusion in ischemic diseases and other conditions in which enhanced tissue oxygenation is needed.

## Methods

**Endothelial cell culture and lentivirus transduction**. Primary cultures of human umbilical vein ECs were obtained from Lonza (#CC-2519) and cultured in complete growth media EGM-2 (Lonza, #CC-3156 and #CC-4176). These cells were transduced at passage 3 with a constitutively active form of R-Ras (R-Ras38V) or

insertless control (mock) using pLenti6 lentivirus expression vector (Thermo-Fisher, K4955-10) to examine the effect of R-Ras signaling in vitro[12]. R-Ras knockdown was carried out by lentivirus transduction of shRNA to target the RRAS sequence (5′-GGAAATACCAGGAACAAGA-3′). The negative control shRNA, which does not target any known sequence of the human, mouse, rat, or zebrafish origin, was obtained from COSMO BIO co., Ltd. (Tokyo, Japan) and cloned into pSIH-H1-puro vector[11]. The complementary DNA (cDNA) for an R-Ras mutant (R-Ras38V D64A), which is constitutively active but incapable of activating PI3K[36], was constructed by PCR. A GSK-3β mutant (GSK-3β S9A) cDNA was obtained from Addgene (plasmid #49492)[50]. These cDNAs were subcloned into pLenti6 vector. In vitro transduction was carried out at one multiplicity of infection per cell.

To determine the role of Akt isoforms in mediating the R-Ras effect, EC were first transduced with R-Ras38V, and 48 h later, either Akt isoform was silenced by transfecting 20 nM of Akt1 short interefering RNA (siRNA) (Sigma, SIHK0094), Akt2 siRNA (Sigma, SIHK0097), or non-targeting siRNA controls (Sigma, SIC001, SIC002) using N-TER™ siRNA nanoparticle transfection system (Sigma, N2788). Likewise, Raptor or Rictor siRNA knockdown or GSK-3β S9A lentivirus transduction was carried out at 48 h after the R-Ras38V transduction.

**In vitro endothelial sprouting assay**. The effect of R-Ras on endothelial morphogenesis was studied in a fibrin gel 3-D culture of ECs[51]. The R-Ras mutant or

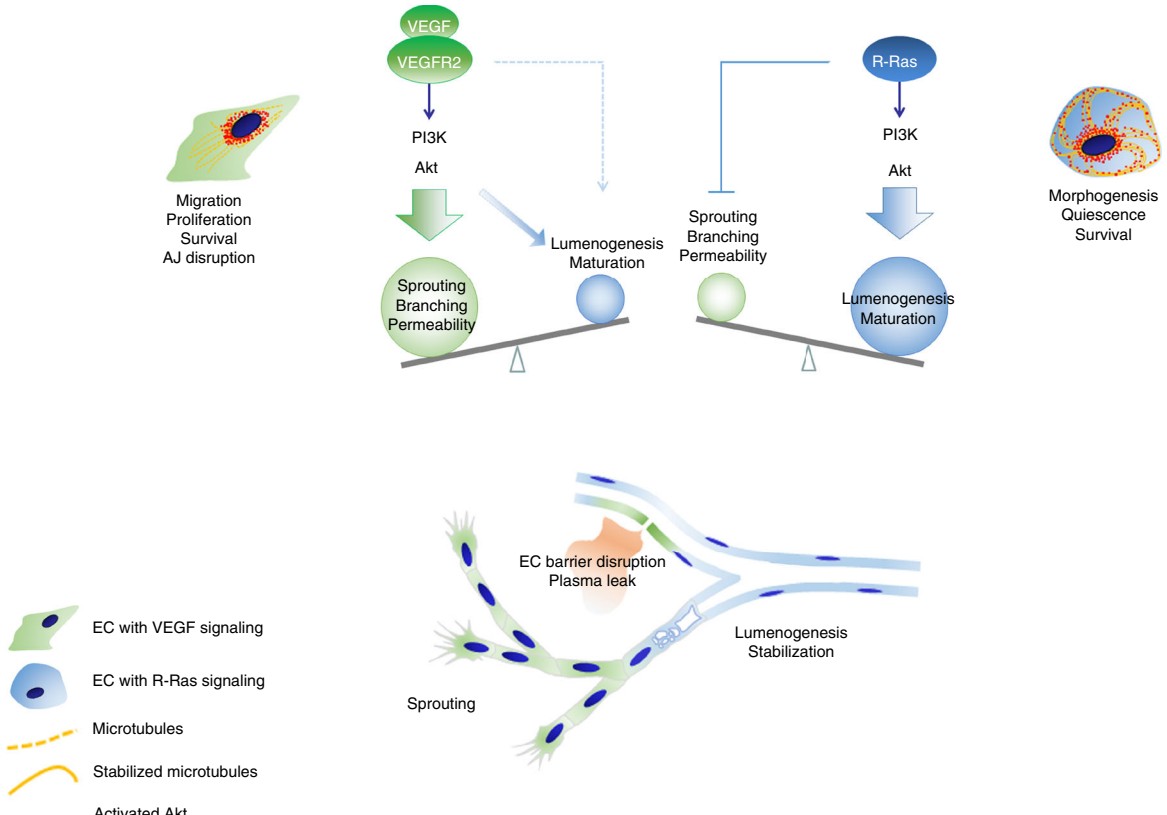

**Fig. 7** Distinct roles of VEGF vs. R-Ras-mediated Akt signaling in angiogenesis. VEGF and R-Ras signaling activate Akt in different manners. Akt accumulates in perinuclear region upon activation by VEGF. In contrast, R-Ras-dependent Akt activation results in the accumulation of activated Akt along the microtubule cytoskeleton as well as at the perinuclear region. The effect of the VEGF-Akt signaling is largely skewed toward vessel sprouting and permeability induction, and it is insufficient for driving vessel lumenization by itself (green). The R-Ras-Akt signaling, on the other hand, promotes lumenogenesis and supports the lumen structure with the stable cytoskeletal architecture of microtubules (blue). R-Ras also limits excessive, nonproductive endothelial sprouting, and block the VEGF-induced disruption of adherens junction (AJ) to maintain endothelial barrier integrity (Sawada et al.[11]). Thus, the VEGF and R-Ras pathways are complementary to each other to generate functional blood vessels for tissue recovery from ischemia

shRNA-transduced ECs were coated on cytodex 3 microcarrier beads (GE Healthcare Life Sciences) at 48 h after transduction. EC-coated beads were then resuspended in a 2 mg/ml fibrinogen solution (Sigma-Aldrich) in PBS at a density of 250 beads per ml. 0.5 ml of the EC-coated bead suspension in fibrinogen solution was added to each well of a four-well chamber slide (Nunc® Lab-Tek® II) containing 0.625 U of thrombin (Sigma-Aldrich). After gels have solidified, 0.8 ml of EGM-2 media containing $2 \times 10^5$ pericytes (human brain microvascular peri-cytes, ScienCell, Inc.) as a feeder layer was added onto the gel in each well. The 3-D culture was maintained in the $CO_2$ incubator at 37 °C for 5–7 days to observe EC sprouting and lumen formation. For counting the number of sprouts, the fibrin gel 3-D culture was fixed and stained with UEA lectin (Vectorlabs, FL-1061). Sprouts were counted at the growing tip of branches.

For Akt silencing in 3-D culture, ECs were transfected with siRNA for either Akt isoform at 48 h after R-Ras38V transduction, and subsequently coated onto microcarrier beads 24 h later. GSK-3β S9A mutant expression in 3-D culture was carried out in a similar time frame after R-Ras38V transduction.

**Confocal 3-D reconstruction of endothelial sprouts.** Confocal z-stacks of endothelial sprouts in 3-D fibrin culture were captured using Nikon A1R confocal microscope. The images of 20 μm in depth and total 40 steps of z-stacks were taken using the same settings of acquisition for the mock and R-Ras38V-transduced EC sprouts. Three-dimensional reconstruction of the confocal image was generated using Volocity® software (PerkinElmer). Snapshots were taken at 1024 × 1024, 300 dpi at selected angles.

**Detergent fractionation of tubulin.** To fractionate soluble and insoluble tubulin, cell lysis buffer (0.5% Triton, 85 mM PIPES [pH 6.9], 1 mM EGTA, 1 mM MgCl₂, 2 M glycerol) was added to cells[52]. The cell lysates were kept at 4 °C for 3 min, and centrifuged for 20 min at 13 000 r.p.m. at 4 °C to yield a supernatant containing soluble (cytosolic) tubulin and a pellet containing insoluble (cytoskeletal) tubulin[52]. α-Tubulin in each fraction was determined by western blot. As a control for microtubule stabilization, cells were treated with paclitaxel 5 ng/ml for 30 min in culture prior to cell lysis.

**Microtubule disruption by nocodazole.** ECs were culture in the fibrin gel 3-D culture system for 5 days to generate endothelial sprouts with lumens. A micro-tubule disrupting agent, nocodazole (Sigma, M1404), was added at 10 μM into the media, and the 3-D culture was incubated for additional 1.5 h. Bright-field microscope images were obtained before and after nocodazole treatment.

**Quantification of lumen size.** Bright-field images of endothelial sprouts in 3-D fibrin gel culture were digitally captured with ×20 magnification. The area size of each hallow structure in the endothelial sprouts (lumen) was determined by morphometry analysis. Sprouts from 10 EC-coated beads in two different culture cells were analyzed for each group. Small vacuoles of <5 μm in length were not included in the analysis.

**Immunofluorescence.** ECs cultured on glass-bottom chamber slides or EC sprouts in 3-D culture were fixed with 2% paraformaldehyde/PBS for 10 min or 30 min, respectively, permeabilized in 0.1% Triton X-100/PBS, and blocked in 10% goat serum/PBS before immunostaining. Cells were stained with α-tubulin (Invitrogen, YOL1/34, diluted 1:500), acetylated-α-tubulin (Lys40) (Cell Signaling Technolo-gies, #5335, diluted 1:250), delta 2-tubulin (Novusbio, NB100-57397, diluted 1:100), detyrosinated α-tubulin (Biolegend, 909503, diluted 1:500), EB1 (Santa Cruz, sc-47704, diluted 1:200), or PODXL (ThermoFisher Scientific, PA5-28116, diluted 1:200) antibody or phalloidin (Invitrogen, A12379, diluted 1:50). ECs and EC sprouts were analyzed by epi-fluorescence (Nikon Eclipse 90i) and confocal (Nikon A1R) microscopy. For quantitative analyses of post-translationally mod-ified and total α-tubulins, the positive area for each immunostaining was deter-mined in each cell by Volocity® software. Total of 15 cells from three different experiments were examined.

**Western blotting analyses.** Cell lysate was prepared from subconfluent mono-layer culture of ECs at 72 h post transduction of R-Ras mutants or control vector. For silencing of an Akt isoform, Raptor, or Rictor, ECs were transfected with siRNA at 48 h after R-Ras38V transduction, and cultured for additional 48 h before

cell lysis. The study with GSK-3β S9A transduction was carried out in a similar time frame after R-Ras38V transduction. For PI3K inhibition, ECs were incubated with LY294002 (25 μM) or DMSO (control) for 1 h before cell lysis.

The following antibodies were used for immunoblotting: rabbit anti-phospho-Akt Ser473 (1:1000, Cell Signaling 4060), rabbit anti-Akt (1:1000, Cell Signaling 9272), anti-phospho-Akt1 Ser473 (1:1000, Cell Signaling 9018), rabbit anti-Akt1 (1:1000, Cell Signaling 2938), rabbit anti-phospho-Akt2 Ser474 (1:1000, Cell Signaling 8599), rabbit anti-Akt2 (1:1000, Cell Signaling 3063), rabbit anti-phospho-GSK-3β Ser9 (1:1000, Cell Signaling 9322), rabbit anti-GSK-3β (1:1000, Cell Signaling 12456), rabbit anti-acetylated-α-tubulin Lys40 (1:1000, Cell Signaling 5335), rabbit anti-α-tubulin (1:1000, Cell Signaling 2125), and rabbit anti-GAPDH (1:10,000, Cell Signaling 2118). Full scans of the western blots shown in Figs. 2–4 are provided in Supplementary Figs. 18 and 19.

**Immunoprecipitation of phospho-Akt.** Cultured ECs were lysed in Triton lysis buffer (1% Triton X-100, 50 mM Tris-HCl, pH 7.4, 2 mM CaCl$_2$, 150 mM NaCl) containing proteinase inhibitor and phosphatase inhibitor cocktails on ice for 10 min. Cell lysate was incubated with anti-phospho-Akt Ser473 (1:200 dilution) overnight at 4 °C, followed by incubation with 120 μl of protein A magnetic beads (CST #8687) for 2 h at 4 °C. Beads were washed and SDS-PAGE was performed, which was followed by anti-Akt1 or Akt2 western blotting for the detection of phosphorylated Akt isoforms. Lysate of 50 μg (5%) was used to indicate the protein input for immunoprecipitation.

**R-Ras activity assay.** Pull-down assays were performed to determine R-Ras activity following the manufacturer's instructions (Cell Signaling, #8821). Briefly, ECs were cultured in low-serum media (2% horse serum without growth factor supplement) for overnight and stimulated with or without 500 ng/ml of recombinant human angiopoietin-1 (R&D Systems) for the indicated times. Cell lysates were prepared and incubated with glutathione resin and GST-Raf1-RBD to pull-down active R-Ras. Western blot analysis of total cell lysate (Input) and the eluted samples (pull-down) were performed using anti-R-Ras antibody (1:2000, Anaspec) and GAPDH antibody (1:10,000, Cell Signaling 2118).

**Animals.** All animal experiments performed here were approved by the Institutional Animal Care and Use Committee at Sanford Burnham Prebys Medical Discovery Institute. The R-Ras knockout mouse line was generated by Lexicon Genetics. The inactivation of *Rras* in these mice is caused by an insertion of the gene-trap vector VICTR20 between exons 4 and 5 of *Rras* on chromosome[12]. These mice have been backcrossed to wild-type mice with the C57BL/6 genetic background >10 times.

**Hindlimb ischemia model.** Femoral artery ligation was performed to induce unilateral hindlimb ischemia in 8-week-old male mice[53,54]. Briefly, left femoral arteries were ligated at two sites at proximal as well as distal to the joint point between profunda and epigastric arteries using polyester 6/0 USP sutures. The contralateral side (right hindlimb) was examined as a normal tissue control.

**Assessment of muscle hypoxia and necrosis.** The level of hypoxia in the GC muscle was analyzed 2 h after intraperitoneal injection of 60 mg/kg pimonidazole into mice at 7 days post ischemia induction. The hypoxic tissues were stained in paraffin sections using Hypoxyprobe kit (Hypoxyprobe, Inc.). Sections were counter stained with hematoxylin to visualize the total muscle area. Hypoxic area was quantified using HALO™ image analysis platform (Indica Labs). The thresholds were set empirically for identifying the area with strong, medium, or weak staining of hypoxic tissues. The area of each level of staining was presented as percent of such area in the total muscle tissue area examined.

For the analysis of muscle degeneration, 5 μm-thick cross-sections were obtained from mid-portion of GC muscles and stained with hematoxylin and eosin. In addition, triphenyltetrazolium chloride (TTC) staining was used to assess the muscle viability[55]. Mice were killed at 14 days after femoral artery ligation, and the GC muscles were collected and cut into 2-mm slices. These slices were incubated for 30 min in 2% TTC (Sigma) solution and fixed for 30 min in 10% (vol/vol) buffered formaldehyde. Viable tissues are stained in red and infarct areas are unstained (pale yellow/white).

**Immunostaining of GC muscles.** Ischemic and contralateral GC muscles were collected from mouse hindlimbs, fixed in 2% paraformaldehyde, dehydrated, embedded in paraffin, and sectioned at 5 μm thickness. After deparaffinization, rehydration, and antigen retrieval (DAKO), sections were blocked and incubated overnight with primary antibodies for CD31 (BD Biosciences) and R-Ras (AbCam). Sections were then washed in PBS with 0.1% Triton X-100 and incubated with Alexa Flour-conjugated secondary antibodies for 1 h at room temperature for fluorescence microscopy. For immunohistochemistry of R-Ras, sections were stained with anti-R-Ras polyclonal antibodies AR43 (gift from Dr. J. Reed) and peroxidase-conjugated secondary antibody (Vector) followed by staining with AEC kit (Life Technologies). For immunofluorescence analyses of frozen sections, GC muscles were snap-frozen in liquid nitrogen. Muscle cross-sections of

10 or 50 μm-thick longitudinal sections were washed in PBS and incubated overnight at 4 °C with primary antibodies diluted in the antibody dilution solution. Sections were then washed in PBS with 0.1% Triton X-100 and incubated with Alexa Flour-conjugated secondary antibodies for 1 h at room temperature. To assess muscle recovery, immunofluorescence was performed with anti-dystrophin antibody (1:200, EMD Millipore) in cryosections of GC muscles, and images were digitally captured (Eclipse 90i, Nikon). The number of dystrophin+ muscle fibers and total muscle fibers in the field (×20 magnification) were manually counted. The muscle recovery was assessed by the dystrophin+ fibers/total fibers (%) ratio determined in non-necrotic area.

**Immunostaining of whole-mounted muscle bundles.** GC muscles were fixed in 2% paraformaldehyde for 1 h at 4 °C and washed three times for 10 min each with PBST (0.2% Triton X-100 in PBS). A small bundle of muscle was carefully dissected by fine forceps from the center portion of the GC muscle. The muscle tissues were then blocked with 1% BSA/PBST (5% normal goat serum) for 1 h at room temperature. The tissues were incubated with rocking overnight at 4 °C with anti-CD31 antibody (1:100 dilution, BD Biosciences, 550274). Next day, tissues were washed overnight in PBS at 4 °C, followed by anti-rat IgG-Alexa 647 conjugate (1:500) at 4 °C for overnight. Immunofluorescence images were acquired using Nikon A1R laser scanning confocal microscope or Nikon 90i fluorescence microscopy.

**BrdU incorporation assay.** BrdU (50 mg/kg) was injected i.p. at 13 days after femoral artery ligation, and the GC muscles were harvested 24 h later. The muscle tissues were snap-frozen in liquid nitrogen. For staining BrdU-labeled cells, frozen sections were incubated with 5 μg/ml proteinase K in PBST for 10 min at room temperature. Sections were then fixed with 1% formaldehyde for 10 min and rinsed in DNase I buffer (50 mM Tris-HCl, 10 mM MgCl$_2$, pH 7.5) for 20 min. The sections were incubated with RNase-free DNase I (0.1 U/μl) in DNase I buffer for 2 h. DNase I was heat-inactivated by incubation at 70 °C for 10 min in preheated 50 mM Tris-HCl, pH 7.5. After blocking nonspecific binding with blocking buffer (10% normal goat serum, 0.2% BSA, 0.3% PBST), the sections were incubated overnight at room temperature with mouse anti-BrdU monoclonal antibody (1:500, ThermoFisher Scientific, MA3-071) and rat anti-mouse CD31 antibody (1:200, BD Biosciences, 550274), followed by incubation with fluorescently labeled secondary antibodies. BrdU and CD31 staining in GC muscles was analyzed by fluorescent microscopy. The percentage of mitotic ECs was determined by: number of BrdU+ nuclei within CD31+ area/total number of nuclei within CD31+ area × 100.

**Vascular area and density.** To assess reparative angiogenesis, a subset of animals (five animals per group) was killed and GC muscles collected at 7 or 14 days after arterial ligation. Cryosections of the GC muscles were incubated with the anti-CD31 antibody (1:200, BD Biosciences, 550274) overnight at 4 °C followed by Alexa Fluor 555 anti-rat immunoglobulin G (1:500; ThermoFisher) for 1 h and analyzed by fluorescence microscopy. The number of CD31+ objects per field (vessel density) and CD31+ area per field (vascular area) were determined in five random fields for each slide using Volocity® image software (PerkinElmer). All data are presented as mean ± SEM.

**Transmission electron microscopy.** GC muscles were collected at 14 days after arterial ligation and cut into small pieces (1–3 mm$^3$), fixed in Karnovsky's fixative (Electron Microscopy Sciences) for 4 h, and washed with PBS two times. Thin sections of GC muscles were prepared and stained with 2% uranyl acetate for 30 s and applied to a continuous carbon grid. The sections were photographed using FEI Morgagni Transmission Electron Microscope system (FEI Company).

**Laser Doppler imaging.** Peripheral blood flow in the mouse foot was analyzed by laser Doppler imaging (PeriScan PIM 3 System, PERIMED). Post-surgical scans were performed immediately after the arterial ligation to confirm the interruption of blood flow in lower limb. Mice were scanned at days 1, 3, 5, 7, and 14. Tissue perfusion was quantified in region of interest defined in the ischemic limb relative to the contralateral, non-ligated side and was displayed as color-coded images. Flow images were then analyzed with blood perfusion imaging software (PIMSoft) and reported as the ratio of flow in the ischemic to non-ischemic hindlimb. Seventeen mice were studied for each group.

**Lectin perfusion.** To histologically visualize the perfusion of newly formed intramuscular capillaries and microvessels, mice were intravenously injected with biotin-labeled tomato lectin (Vector) at 14 days post ischemia induction. Mice were killed 5 min later and GC muscles were isolated. A small piece (100 μm thickness) from the center portion of the GC muscle was carefully dissected and fixed in 2% paraformaldehyde. Small bundles of GC muscles were doubly stained in whole-mount for CD31 and biotinylated lectin to visualize all vessels (perfused and non-perfused) and perfused vessels, respectively. The images of 10 random areas in each specimen were captured, five mice per group. The perfusion efficiency was calculated as a ratio of perfused vessel area (CD31/lectin double positive area) to total vascular area (CD31 positive area) by Volocity® image software (PerkinElmer).

**Endothelial cell-targeted in vivo gene transfer**. A lentiviral expression vector, pLenti6/*Cdh5*-R-Ras38V, was generated for EC-specific expression of R-Ras38V in mouse tissues. This construct was generated by replacing CMV promoter in the original pLenti6 vector with the mouse VE-cadherin (*Cdh5*) promoter sequence[35]. A similar expression vector was constructed for R-Ras38V D64A mutant expression. A lentiviral vector with *Cdh5* promoter without cDNA insert was also generated as a control.

For R-Ras KO rescue experiments, hindlimb ischemia was induced in the R-Ras KO mice, and 3 days later, mice were randomly assigned into two groups. The lentivirus harboring the pLenti6/*Cdh5*-R-Ras38V expression vector or control vector was intramuscularly injected into the GC muscle at three injection sites, 10 μl each, $1.2 \times 10^7$ TU total using a Hamilton syringe. At day 14, mice were intravenously perfused with biotinylated tomato lectin through the tail vein 5 min before resecting the GC muscles for analysis. Five mice per group were examined. To confirm the induction of R-Ras38V expression, total tissue RNA was isolated from the GC muscle using RNeasy Mini Kit (Qiagen, 74134). cDNA was synthesized using a high capacity cDNA reverse transcription Kit (Applied Biosystems, 4368814). Reverse transcription PCR (RT-PCR) was carried out using a primer set for human *RRAS*: 5′-GACCCCACTATTGAGGACTCC-3′ and 5′-CTACAGGAGGACGCAGGG-3′. A primer set for mouse *Ppia* (Integrated DNA Technologies) was used to standardize RT-PCR.

**Akt inhibitor treatment**. A set of mice that received pLenti6/*Cdh5*-R-Ras38V lentivirus injection into GC muscles were treated with an Akt inhibitor MK-2206. MK-2206 (Apexbio Technology LLC) was dissolved in 30% captisol® (Ligand Pharmaceuticals, Inc.) and administered p.o. at 120 mg/kg by gavage every 2 days[37] starting from 1 day after the lentivirus injection (day 4 post femoral artery ligation) for total five times. At day 14, 100 μl of 1 mg/ml tomato lectin was injected i.v. for vessel perfusion analysis, and mice were killed to collect GC muscles. A control group received vehicle alone via gavage. Three animals were examined per group.

**Endothelial cell isolation from GC muscle**. GC muscles were resected at 14 days post ischemia induction and dissociated into single cell suspension by enzymatic digestion with collagenase II (500 U/ml, Sigma #6885), collagenase D (1.5 U/ml, Roche #11088882001), and dispase II (2.4 U/ml, Roche, #04942078001). Endothelial cells were sorted from the cell suspension using anti-mouse CD31 MACS magnetic microbeads and MS column (Miltenyi Biotec). Endothelial cell purity was assessed by VE-Cadherin expression and LDL uptake assay (Biomedical Technologies Inc., BT-902). The isolated cells were lysed and used for western blot analyses to determine R-Ras expression, Akt and GSK-3β phosphorylation, and tubulin acetylation.

**Adenovirus-mediated VEGF gene therapy**. Adenovirus carrying mouse VEGF-A$_{164}$ (Cell Biolabs Inc.) was injected at $3 \times 10^8$ p.f.u. ($1 \times 10^8$ p.f.u./10 μl × 3 injections) into the calf (GC) muscles of wild-type mice immediately after femoral artery ligation. The GC muscles were collected for immunofluorescence analyses 11 days later. Frozen sections of 50 μm were used for lectin perfusion assay and PODXL staining. Frozen sections of 10 μm were used to detect R-Ras expression.

**Statistical analysis**. Statistical analyses were performed using the two-tailed Student's *t* test to compare two experimental groups and the one-way analysis of variance with Tukey's multiple comparison test for multiple experimental groups. Calculations were performed using GraphPad Prism v7.00 software. $P < 0.05$ was considered significant. Error bars represent the SEM. Data on the graphs are presented as mean ± SEM.

**Data availability**. All data relevant to this study are presented in the manuscript's main figures or the Supplementary files and available from the authors upon request.

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

## Acknowledgements

We thank Drs. W. Stallcup of SBP Discovery, M. Fukuda, and M. Nonaka of National Institute of Advanced Industrial Science and Technology for helpful comments on the manuscript. We also thank Dr. T. Urakami and Ms. A. Urakami for their assistance on mouse femoral ligation model and other experiments. Histological preparations were done at the Histology and Imaging core facilities. Molecular cloning of expression vectors was done by Protein Production core facility. This work was supported by National Cancer Institute Grant CA125255, National Science Foundation Grant CBET-1403535, American Heart Association fellowship 15POST25700319 to F.L., Bankhead-Coley Caner Research Program 4BB17, and Florida Breast Cancer Foundation Grant.

## Author contributions

F.L. initiated the in vitro studies of endothelial lumenogenesis, R-Ras-Akt signaling, and microtubule stabilization. F.L. and M.K. designed the experiments. F.L. and J.S. performed the in vitro studies. F.L. conducted the in vivo studies with mouse models. F.L. and M.K. wrote the paper.

## Additional information

**Competing interests:** The authors declare no competing financial interests.

