## [Peer Review File · Nature Communications]

Reviewers' comments:

Reviewer #1 Remarks to the Author:

This is an excellent report demonstrating a role for R-Ras in vessel lumenization mediated by signaling through Akt/GSK3beta and microtubules. The report provides a number of other important observations, most notably that VEGF-mediated Akt activation signals through a divergent but as yet unclear mechanism that involves downregulation of R-Ras in ECs, which may partly explain the failure of VEGF-mediated pro-angiogenic therapies. Overall, the studies are well performed and support the authors' conclusions. In general, my comments are minor with several exceptions. As the authors point out, the mechanism of the divergent Akt signaling is still not clear. While this may be beyond the scope of the current report, I believe there are some modest experiments (outlined below) that could shed additional light on this difference. Moreover, in some cases the paper lacks quantitative and functional analyses, which would strengthen the paper. Specific comments are detailed below.

Major comments:

1. Upstream activation of R-Ras - The lack of R-Ras activation by VEGF despite its activation of Akt raises the question of whether other factors, most notably Angiopoietin-1, might activate R-Ras to promote lumenization. Ang-1 promotes EC migration and sprouting and has been linked to venous enlargement, which may be Akt-mediated, and it is well known to promote vessel maturation (i.e., inhibit leakiness), thus it appears to fit quite well into the right side of Figure 7. The authors should test Ang-1's effect on R-Ras and lumenization.
2. Time course of R-Ras activation - The authors raise an important question, i.e., whether the duration of Akt activation is responsible for the difference in VEGF-mediated vs. R-Ras38V-mediated effects (p. 12, lines 312-313). However, one wonders whether this is simply related to constant expression of active R-Ras vs. transient stimulation with VEGF-A. Do the authors know of a way to transiently activate R-Ras? As in comment #1, perhaps a way to test this is with Ang-1 stimulation. Otherwise, it is impossible to know the real reason for the differential effects on microtubules in Fig 3.
3. Quantification of effects of HLI - The authors have shown effects on perfusion primarily through infusion of lectin, which shows functional vessels, although they have also included LDPI in Suppl Fig 6 as a functional measure of blood flow. Although this actually measures blood flow velocity, it is widely used in such studies and provides a reliable comparison to other studies. In this regard, it would be helpful to see LDPI in some of the other studies, for example, in Suppl Fig 10 and in Fig 5 after EC-specific R-Ras delivery to know the overall effects on perfusion. In the case of the AdVEGF studies, it has been shown repeatedly that VEGF delivery improves perfusion, at least transiently, thus it is somewhat surprising that the neovessels appear largely non-functional, and it would be important to compare them to controls by this standard measurement. Similarly, does the EC-specific R-Ras delivery affect functional perfusion aside from that shown in panels c and d?
4. Quantification of effects on muscle injury - In Fig 4g, the authors should quantify TTC staining as a measure of muscle injury, and quantification of injury should also be done in Fig 5f. Alternatives to injured area might include markers of muscle regeneration, which would be dependent on an intact vascular supply, including centralized myonuclei, intact dystrophin staining, or staining for embryonic myosin heavy chain.

Minor comments:

1. Is R-Ras expressed in monocytes, which are known to play a role in angiogenesis and arteriogenesis?

2. Suppl Fig 7 - Are the authors arguing that there is muscle atrophy? If so, they should measure myofiber cross-sectional area
3. Suppl Fig 8 - Are animals #1-2 controls? This is not indicated in the figure legend. In panel b, is the antibody specific for human R-Ras? If not, why is there no expression in mock-transduced cells?
4. Suppl Fig 9b - the data on microtubule stabilization are not particularly robust, unlike those in Fig 1d and e or the effects on pAkt
5. Suppl Fig 10 - What strain were the mice? The legend simply says wild-type. Are these C57BL/6?

Reviewer #2 Remarks to the Author:

In this manuscript, the authors communicate findings that bring additional clarity to the role of R-Ras in endothelial lumen formation. In particular, the authors demonstrate that R-Ras has a profound effect in microtubule organization through a mechanism that requires Akt activation. Furthermore they showed that the effect of VEGF and R-Ras in activation of Akt provide very distinct outcomes.

Overall I found the data compelling and well presented. While the contribution of R-Ras to vessel formation has been already partially explored, the present study offers a more critical analysis of the effect of R-Ras and expands a role of Akt in lumen formation, explaining some of the previous phenotypes of transgenic overexpression of Akt in vivo. Naturally, the findings also open a plethora of additional questions, but in my opinion the study is substantial and it advances our understanding of lumen formation.

A few important concerns:

- 1) The authors use the term "stability of microtubules" however this was never really shown directly. So either rephrase or demonstrate that indeed microtubules are more stable by doing live imaging using FRAP or a similar technique.
- 2) In relation to some panels in Figure 1, the authors make comments related to polarity. It will be important to use gamma tubulin or a Golgi marker to more clearly identify the polarity of those cells.
- 3) Is acetylation and detyrosination of alpha tubulin altered upon blockade of Akt activation? In other words, does R-Ras affect microtubule post-translational modifications in a manner that is independent or dependent of Akt?
- 4) What is the contribution of delta 2-tubulin in the process of lumen formation induced by R-Ras? What happens if delta 2-tubulin is silenced?
- 5) While the data on R-Ras on microtubules is convincing, one can only ponder whether these effects do take place in vivo, as most of the results on microtubule modifications are shown using a powerful constitutively active form of R-Ras. Can the authors show activation of R-Ras in endothelial cells as a lumen is being formed in an incipient sprout under physiological conditions? While the KO studies are compelling, there is no evaluation of microtubules in those vessels. Provision of these data would most certainly help.
- 6) The introduction of VEGF is interesting, but incomplete, it does bring more questions than answers (perhaps removing it might be best). For example: is the only difference between the effect of VEGF and R-Ras the length of the signaling effect? If so... what happens under physiological conditions (?), as R-Ras would not be constitutively active. Is the difference between VEGF and R-Ras physiological and real or an artifact related to the reagents used? The same question relates to the localization of Akt.
- 7) Does constitutive active Akt result in the same phenotype/outcome on microtubules? Again, is

R-Ras acting exclusively through Akt or are other players involved downstream R-Ras?

8) The *in vivo* data is nice, but the use of PODXL as a mean to visualize lumenized vessels not as convincing. A combination of intravascular lectins and PECAM should be included to ensure that indeed lumen is compromised. Alternatively, vasculature in skeletal muscle can be very nicely observed *in vivo* by videomicroscopy of fluorescent endothelial cells with fluorescently labeled blood cells (this can be an alternative).

9) As stated previously, it will be important to show the status of microtubules in the KO mice.

Reviewer #3 Remarks to the Author:

In their manuscript Li and colleagues provide evidence that in endothelial cells (ECs) R-Ras signals through phosphoinositide 3-kinase (PI3K)-Akt/GSK-3 β to stabilize microtubules (MTs). Authors go on suggesting that the R-Ras/PI3K/Akt/GSK-3 β /MT stabilization signaling pathway would promote lumenization of EC tubes *in vitro* and blood vessels *in vivo*. Authors also propose that the modes of activation of the PI3K/Akt/GSK-3 β signaling pathway and the ensuing modulation of MT dynamics by VEGF and R-Ras are distinct. With the only exception of the potential link between R-Ras and lumenogenesis, this is largely a confirmatory manuscript. Furthermore, the hypothetical link between R-Ras and lumen formation is not robustly and convincingly addressed neither *in vitro* nor *in vivo*. A major issue is also represented by the fact that most data are described in a qualitative (e.g. using adjectives such as pronounced, considerable, considerably, or significantly) and not in a quantitative way (i.e. by means of quantitative parameters and numbers coupled to statistical analysis).

Specific comments

1. It is already known that: i) R-Ras activates PI3K (Marte et al., 1997, *Curr Biol* 7, 63-70); ii) PI3K signals through Akt (Onishi et al., *Genes Cells*, 2007, 12:535-546) and GSK-3 β (Zumbrunn et al., 2001, *Curr. Biol.*, 11:44-49) to stabilize MTs; iii) microtubule assembly and tubulin modifications control EC lumen formation (Kim et al., *Blood*. 2013, 121:3521-3530). Therefore, it is not so innovative to describe a link between the R-Ras/PI3K/Akt/GSK-3 β /MT stabilization signaling and endothelial lumenogenesis.

2. In figures 1 and 2, the analysis of EC morphogenesis *in vitro* is not careful and no robust conclusion can be drawn about the possible role of R-Ras in the lumenization of endothelial sprouts. Phase contrast microscopy coupled to fluorescence microscopy, in the absence of a detailed confocal analysis along EC Z-axis of well-known apical domain markers, such as podocalyxin (PODXL; Horvat et al., *J. Cell Biol.*, 1986, 102:484-491), do not formally demonstrate that structures lined by ECs in 3D fibrin are hollow tubes. As far as they appear, the hypothetical lumens may for example just be bent chains formed by ECs. Electron microscopy coupled to PODXL immunogold labeling (Horvat et al., *J. Cell Biol.*, 1986, 102:484-491) would be extremely helpful here.

3. In lines 82-86, authors write "Many of these sprouts showed that they have begun forming lumens (Figure. 1b). However, these lumens are disconnected, and the sprouts are incompletely lumenized (Figure. 1b). The expression of constitutively active R-Ras (R-Ras38V) dramatically enhanced the EC lumenogenesis to create well-defined, uninterrupted lumens (Figures. 1a, b)." Supposing that authors provide a formal and clear demonstration that R-Ras controls lumen formation, afterwards lumenogenesis must be thoroughly quantified in basal conditions as well as after R-Ras38V overexpression and endogenous R-Ras knock-down. Furthermore, the ability of exogenously-delivered wild type or mutant (e.g. Akt-independent D64A) R-Ras constructs to rescue the disruption of lumenogenesis caused by R-Ras silencing should be verified.

4. In lines 99-106, authors write "The microtubule stabilizing effect of R-Ras was demonstrated by

markedly increased acetylation and detyrosination of α -tubulin (Figures. 1c,d, Supplementary Figure. 2a) as well as pronounced elongation of microtubules that reach the cell membrane (Figure. 1f, Supplementary Figure. 2b). Moreover, a considerable accumulation of delta 2-tubulin was evident (Figure. 1e) indicating the formation of long-lasting, highly stable microtubule cytoskeleton network in these cells. The R-Ras-dependent microtubule stabilization was also observed in endothelial sprouts in the 3-D culture that exhibited enhanced lumenogenesis and uninterrupted capillary-like tubular structures (Figure. 1g)."

All these effects must be quantified and not merely described. Which is the quantitative extent of modification of α -tubulin acetylation and detyrosination? How MT elongation was quantitatively defined? In how many ECs was MT elongation quantified? Statistical analysis must be provided as well.

5. In lines 82-86, authors write "In the lumenized sprouts, microtubules elongated from the microtubule organizing center positioned at the apical side of the nucleus in each cell, indicating the polarization of ECs (Figure. 1h)."

Neither an apical domain nor a microtubule organizing center marker was stained to support authors' statements. Furthermore, it appears that microtubules belonging to different opposite ECs get in contact: this would suggest that, differently from what hypothesized by authors, endothelial tubes of R-Ras38V-overexpressing ECs do not contain any lumen, but their structure is just due to a different 3D organization of ECs and their MTs, compared to tubes formed by control ECs.

6. In Figure 3, it is not correct to compare the activation of the pAkt/pGSK-3 β /Ac- α -tubulin pathway caused by chronic R-Ras38V overexpression and acute VEGF-A stimulation. The correct experiment would be to compare pAkt/pGSK-3 β /Ac- α -tubulin activation in control and R-Ras silenced ECs that were stimulated or not with VEGF-A. In such a way, authors would understand if R-Ras is involved or not in the transient pAkt/pGSK-3 β /Ac- α -tubulin activation that is elicited by acute VEGF-A stimulation.

7. In lines 156-157, authors write "The immunofluorescence staining of ECs indicated that the Ser 473-phosphorylated Akt localizes at the perinuclear late endosomes upon cell stimulation with VEGF-A (Figure. 3c)".

No late endosome marker is provided to support this statement.

8. Speaking about revascularization of ischemic skeletal muscles in R-Ras KO mice, in lines 182-184, authors write "However, the immunoreactivity for podocalyxin (PODXL), which highlights endothelial lumen, was frequently absent in these vessels indicating a deficiency in lumenogenesis (Figure. 4a)."

Authors must take into account that the acquisition of EC apicobasal polarity usually relies on the delivery of PODXL-containing vesicles to the future apical plasma membrane, rather on the modulation of PODXL expression (for review see Zeeb et al., Curr. Opin. Cell Biol., 2010, 22:626-632). In ECs, is R-Ras controlling PODXL protein levels rather apical delivery from endosomal stores? If yes, is R-Ras acting on PODXL protein at transcriptional, translation, or degradative level? These aspects may also be easily addressed in cultured ECs. PODXL immunogold labeling electron microscopy on muscle sections of R-Ras wild type and KO mice would be also extremely helpful.

9. Most importantly, since R-Ras is an effective activator of anti-apoptotic PI3K-Akt signaling, the impaired blood vessel re-perfusion observed in the muscle of R-Ras KO mice may be due EC apoptosis. Author must hence disprove the fact that lack of R-Ras may significantly increase EC apoptosis elicited by stressful events, such as femoral artery ligation. EC survival rather than blood vessel lumenogenesis may be the mechanism by which R-Ras influences post-natal angiogenesis

10. Scale bars are lacking in Figure panels 1d, 1e, 1f, 1i, 2d, 2h, 2f, 2i, 3a, 3c, 6b, S1a, S2a, S2b, S3b, and S9b.

11. Line 367-368: "The cDNA for an R-Ras mutant (R-Ras38V D64A), which is constitutively active but incapable of activating PI3K (ref.3)".
The correct reference is not ref. 3, but ref. 33.

Response to Reviewers

We thank for the encouragement for this manuscript and recognition of the significance of our findings by the reviewers. The followings are responses to specific questions and concerns.

Reviewer 1

Major comments

1. Upstream activation of R-Ras - The lack of R-Ras activation by VEGF despite its activation of Akt raises the question of whether other factors, most notably Angiopoietin-1, might activate R-Ras to promote lumenization. Ang-1 promotes EC migration and sprouting and has been linked to venous enlargement, which may be Akt-mediated, and it is well known to promote vessel maturation (i.e., inhibit leakiness), thus it appears to fit quite well into the right side of Figure 7. The authors should test Ang-1's effect on R-Ras and lumenization.

We examined the effect of Ang-1 on R-Ras activity in ECs in relation to Akt activation, GSK-3 β phosphorylation, and tubulin acetylation. As expected, Akt was sharply but transiently activated by Ang-1 with a peak activity at 15 min. The phosphorylation of GSK-3 β also peaked at 15 min, which was accompanied by transient acetylation of α -tubulin (Supplementary Fig. 7a). Ang-1 moderately increased R-Ras activity (Supplementary Fig. 7b). However, the temporal pattern of R-Ras activation does not seem to match the pattern of Akt/GSK-3 β phosphorylation by Ang-1 that declines sharply after 15 min. R-Ras activation by Ang-1 may be secondary to cellular responses to this growth factor via mechanisms such as alterations of cytoskeleton or cell adhesion. More importantly, Ang-1 did not enhance the EC lumen formation activity in the 3-D culture analysis (Supplementary Fig. 7c, d). Taken together, these results do not seem to support a role of Ang-1 in promoting endothelial lumenogenesis. We included these results in the revised manuscript.

2. Time course of R-Ras activation - The authors raise an important question, i.e., whether the duration of Akt activation is responsible for the difference in VEGF-mediated vs. R-Ras38V-mediated effects. However, one wonders whether this is simply related to constant expression of active R-Ras vs. transient stimulation with VEGF-A. Do the authors know of a way to transiently activate R-Ras? As in comment #1, perhaps a way to test this is with Ang-1 stimulation. Otherwise, it is impossible to know the real reason for the differential effects on microtubules in Fig 3.

The observed microtubule stabilization and enhanced lumen formation cannot be merely due to an artifact of the R-Ras38V expression system we used. We repeatedly demonstrated the importance of endogenous R-Ras for these activities in the *in vitro* (Fig. 1, Fig. 2b, Fig. 3c, Supplementary Fig. 1c, 3a-c, 4, 8) and *in vivo* (Fig. 4) studies. The silencing or knockout of endogenous R-Ras reduced the level of Akt activity and elevated that of GSK-3 β in cultured ECs or ECs of intact vessels. Considering the known function of R-Ras in vessel wall stability, it is conceivable that the basal level of endogenous R-Ras activity may be relatively high in ECs providing constant signaling. The maintenance of stable microtubule network for stable lumen structure would also require constant activity. In agreement with this notion, there is a sizable contribution of endogenous R-Ras to the basal level of Akt activity (pAkt), GSK-3 β phosphorylation, and α -

tubulin acetylation in ECs without growth factor stimulation in low-serum culture (Fig. 3c, R-Ras-silenced cells/no VEGF vs. control cells/no VEGF). At this moment, there is no definitive answer to the signaling duration question, and further investigations on this topic is warranted for future studies. In this manuscript, we proposed the signaling duration as one possible explanation for the differential Akt effects and discussed this possibility in the *Discussion* section. We also reemphasized the importance of endogenous activity of R-Ras for Akt signaling, microtubule stability, and EC lumenogenesis throughout the manuscript.

3. Quantification of effects of HLI - The authors have shown effects on perfusion primarily through infusion of lectin, which shows functional vessels, although they have also included LDPI in Suppl Fig 6 as a functional measure of blood flow. Although this actually measures blood flow velocity, it is widely used in such studies and provides a reliable comparison to other studies. In this regard, it would be helpful to see LDPI in some of the other studies, for example, in Suppl Fig 10 (new S15) and in Fig 5 after EC-specific R-Ras delivery to know the overall effects on perfusion. In the case of the Ad-VEGF studies (new Suppl Fig. 15), it has been shown repeatedly that VEGF delivery improves perfusion, at least transiently, thus it is somewhat surprising that the neovessels appear largely nonfunctional. Similarly, does the EC-specific R-Ras delivery affect functional perfusion aside from that shown in panels c and d?

The tissue depth limitation for the laser Doppler measurement is only about 1 mm for most tissues, and importantly, most of the signal come from blood flow in the skin/subcutaneous circulation rather than muscles. However, we inject R-Ras lentivirus directly into the GC muscle 2~3 mm deep from the surface for this study. The reported success of laser Doppler analysis for intramuscularly injected Ad-VEGF is likely due to the fact that it is an overproduced growth factor, which penetrates from the muscle to skin. We have consulted with PeriMed, the manufacture of the instrument, and after these considerations, we concluded that the most accurate method for assessing intramuscular microvessel perfusion is the analysis with lectin perfusion. This is a direct and reliable quantitative measurement to evaluate the perfusion of each vessel, and we analyze this in whole-mounted GC muscle rather than sections so that we can visualize the intact vessel network structure as a whole within the specimen. The disadvantage of this method is that it is not real-time measurement of live animals, which is not an issue for our study.

To accompany this vessel function analysis, we included the quantification of dystrophin-positive GC muscle recovery after EC-specific R-Ras lentivirus injection (Fig. 5g) to corroborate the observed increased lumen formation and perfusion recovery.

As this reviewer points out, it is well known that VEGF increases overall blood flow in the muscle. This can be due to either altered vessel number or altered perfusion of these vessels or both. The degree of muscle perfusion is a net effect of these two factors. Because we are investigating endothelial lumenogenesis, we focused on the vessel function itself in this particular analysis. Therefore, we analyzed the perfusion of each vessel (lectin perfusion efficiency %). In this analysis, we found that, when robust angiogenesis is forced by VEGF therapy, new vessels have a reduced capacity for lumenogenesis and therefore perfusion. This doesn't mean overall amount of blood flow into the muscle got reduced since the overall vascularization increased by the VEGF therapy. We clarified this point by revising the label on the graph as "Lectin perfusion of vessels (%)" in the figure.

4. Quantification of effects on muscle injury - In Fig 4, the authors should quantify TTC staining as a measure of muscle injury, and quantification of injury should also be done in Fig 5. Alternatives to injured area might include markers of muscle regeneration, which would be dependent on an intact vascular supply, including centralized myonuclei, intact dystrophin staining, or staining for embryonic myosin heavy chain.

We have added analyses of dystrophin immunostaining to quantify the muscle injury and amount of functional muscles (Fig. 4i, 5g).

Minor comments

1. Is R-Ras expressed in monocytes, which are known to play a role in angiogenesis and arteriogenesis?

Tissue immunostaining indicates that strong expression of R-Ras is mainly found in ECs, pericytes, and smooth muscle cells. However, as we discussed in the Discussion section, we cannot rule out a potential contribution of R-Ras expressed by other cell types at low levels. For this reason, we engineered a VE-cadherin (*cdh5*) promoter-driven R-Ras expression lentivirus construct for EC-specific rescue of the R-Ras global KO phenotype. This study demonstrated that R-Ras signaling in ECs alone promotes endothelial lumenogenesis and improve vessel perfusion, which are blocked by Akt inhibition (Fig. 5 and 6). These *in vivo* observations corroborated the *in vitro* findings (Fig. 1, 2, Supplementary Fig. 1 – 6, 8) to show the significance of endothelial R-Ras signaling for EC lumenogenesis.

2. New Suppl Fig 12 - Are the authors arguing that there is muscle atrophy? If so, they should measure myofiber cross-sectional area

We included a new figure (Suppl. Fig. 12b) in the revised manuscript to compare cross-sectional area of muscle fibers.

3. New Suppl Fig . 13 - Are animals #1-2 controls? This is not indicated in the figure legend. In panel b, is the antibody specific for human R-Ras? If not, why is there no expression in mock-transduced cells?

The mice used in this *RRAS* gene transfer validation study were all R-Ras KO mice. Only the left legs (GC muscles) of Animal # 3 and 4 (indicated by red L) received lentivirus injection for human *RRAS* gene *in vivo* transduction (Suppl. Fig. 13a). The rest are all control legs, which did not receive virus injections (1R and L, 2R and L, 3R, and 4R). We revised the figure and figure legend to clarify. The western blot does not detect any R-Ras (human or mouse) in the mock virus-injected control group because the recipient mice are R-Ras KO (Suppl. Fig. 13b).

4. New Suppl Fig 14b - the data on microtubule stabilization are not particularly robust, unlike those in Fig 1d and e or the effects on pAkt.

That is correct. R-Ras38V D64A mutant used in this study is constitutively active like R-Ras38V but cannot activate Akt due to the D64A mutation (Suppl. Fig. 14a). As a result, the expression of

this mutant fails to stabilize microtubules in ECs (Suppl. Fig. 14b). This experiment is to provide an additional evidence to support the significance of Akt for the R-Ras effect.

5. New Suppl Fig 15 - What strain were the mice? The legend simply says wild-type. Are these C57BL/6?

All mice used in this study are in the C57BL/6 background. The R-Ras KO line has been maintained by backcrossing to this strain >20 times. We revised the figure legends to clarify.

Reviewer 2

1. The authors use the term “stability of microtubules” however this was never really shown directly. So either rephrase or demonstrate that indeed microtubules are more stable by doing live imaging using FRAP or a similar technique.

The three microtubule stability markers, α -tubulin acetylation, de-tyrosination, and delta-2 tubulin that we used in this study, are well-established and commonly used in published studies to assess microtubule stability. Particularly, the accumulation of delta-2 tubulin (i.e. post-translationally modified α -tubulin, from which C-terminal tyrosine and glutamic acid residues are both removed) indicates long-lived highly stable microtubules (*Paturle-Lafanechère et al., 1994; Janke and Bulinski 2011*). We believe that it is reasonably acceptable to use these markers as a surrogate method to assess microtubule stability.

Furthermore, we conducted a biochemical analysis as additional confirmation. The accumulations of α -tubulin in the detergent-soluble and insoluble fractions were determined for the control, R-Ras-transduced, or silenced ECs. In this study, taxol, a strong microtubule-stabilizing agent, was used as a positive control. The treatment of ECs with taxol resulted in significant accumulation of α -tubulin in the detergent-insoluble fraction due to the increase of stable microtubule cytoskeleton in the cells accumulated in this fraction. Similarly to the taxol treatment, the expression of R-Ras38V in ECs caused α -tubulin accumulation in the detergent-insoluble fraction. On the other hand, the silencing of endogenous R-Ras by shRNA increased α -tubulin in the soluble fraction and decreased in the insoluble fraction. These results corroborate the results of microtubule stability-marker staining analyses and further support the role of R-Ras in the formation of stable microtubule cytoskeleton.

Reference

Accumulation of delta 2-tubulin, a major tubulin variant that cannot be tyrosinated, in neuronal tissues and in stable microtubule assemblies. Paturle-Lafanechère et al. Journal of Cell Science 1994. 107, 1529-1543.

Post-translational regulation of the microtubule cytoskeleton: mechanisms and functions. Janke and Bulinski. Nat Rev Mol Cell Biol. 2011. 12(12):773-786.

2. In relation to some panels in Figure 1, the authors make comments related to polarity. It will be important to use gamma tubulin or a Golgi marker to more clearly identify the polarity of those cells.

We now included γ -tubulin staining of EC sprouts to indicate microtubule organizing center and cell polarity in the revised manuscript (Supplementary Fig. 5).

3. Is acetylation and detyrosination of alpha tubulin altered upon blockade of Akt activation? In other words, does R-Ras affect microtubule post-translational modifications in a manner that is independent or dependent of Akt?

We demonstrated the Akt-dependence of the R-Ras effect by both *in vitro* (Fig. 2) and *in vivo* (Fig. 6) analyses. The silencing of Akt1 or Akt2 isoform blocked the effect of R-Ras on tubulin acetylation and lumen formation (Fig. 2g, h, i). The inhibition of Akt upstream, PI3K or Rictor, by LY294002 or Rictor knockdown also blocked the R-Ras effect (Fig. 2c, d). Furthermore, we demonstrated in the R-Ras KO mouse rescue experiments that the inability to activate Akt (either by the use of R-Ras38VD64A mutant or Akt inhibitor) blocks the effect of EC-specific R-Ras38V transduction on the tubulin acetylation *in vivo* (Fig. 6b). These results corroborated the effects of Akt blockade on the lumen formation and lectin perfusion in the ischemic GC muscles (Fig. 6c, d).

4. What is the contribution of delta 2-tubulin in the process of lumen formation induced by R-Ras? What happens if delta 2-tubulin is silenced?

As discussed above, delta-2 tubulin is a post-translationally modified α -tubulin that is found in highly stable microtubules. The C-terminal tyrosine gets enzymatically removed from α -tubulin followed by the removal of the second residue (glutamic acid). The presence of these two modifications in α -tubulin indicates long-lived stable microtubules (*Paturle-Lafanechère et al., 1994; Janke and Bulinski 2011*). Thus, our results support the importance of microtubule stabilization for endothelial lumenogenesis.

5. While the data on R-Ras on microtubules is convincing, one can only ponder whether these effects do take place in vivo, as most of the results on microtubule modifications are shown using a powerful constitutively active form of R-Ras. Can the authors show activation of R-Ras in endothelial cells as a lumen is being formed in an incipient sprout under physiological conditions? While the KO studies are compelling, there is no evaluation of microtubules in those vessels. Provision of these data would most certainly help.

We analyzed microtubules in the ECs of intact blood vessels in ischemic GC muscle by immunostaining (Fig. 4d, 6b) and by western blot of isolated ECs from these vessels (Fig. 4c). Both analyses showed impaired α -tubulin acetylation as a result of disrupting endogenous R-Ras. These results are consistent with the *in vitro* R-Ras silencing studies, which demonstrated the necessity of endogenous R-Ras for the post-translational modifications of microtubules (Fig. 1c-g, Supplementary Fig. 3a, b) and for EC lumenogenesis in 3D culture (Fig. 1a, b, Supplementary 1c). The combined results argue that the observed effects of R-Ras is not an artifact of expressing constitutively active R-Ras and support the physiological importance of this small GTPase in microtubule regulation and lumenogenesis.

6. The introduction of VEGF is interesting, but incomplete, it does bring more questions than answers (perhaps removing it might be best). For example: is the only difference

between the effect of VEGF and R-Ras the length of the signaling effect? If so... what happens under physiological conditions (?), as R-Ras would not be constitutively active. Is the difference between VEGF and R-Ras physiological and real or an artifact related to the reagents used? The same question relates to the localization of Akt.

Our result indicates that VEGF alone is insufficient to support microtubule stabilization or to promote endothelial lumenogenesis despite its ability to strongly activate Akt and inhibit GSK-3 β for a short time. We think that this finding itself is a novel and important observation worthy to report as VEGF is a primary growth factor to induce angiogenesis. It is expected that such a short effect would not contribute to the stability of the microtubule network required for the lumen formation and maintenance since sustained GSK-3 β inhibition is necessary for such a stability. As we mentioned in the response to Reviewer 1 Question #2, there is a sizable contribution of endogenous R-Ras to the basal levels of Akt activity, GSK-3 β phosphorylation, and α -tubulin acetylation in ECs without growth factor stimulation. These observations support an idea that a relatively high basal activity of endogenous R-Ras may be constantly present in ECs. Since the observed effects are results of disrupting endogenous R-Ras, our results demonstrate a normal physiological function of R-Ras. We proposed signaling duration as one of the possible explanations for the differential Akt signaling effects in the *Discussion* section.

Our results also suggests a difference in the localization of activated Akt. In addition to the immunofluorescence localization analysis of activated Akt (Fig. 3d), we performed a biochemical analysis. This study showed that R-Ras signaling leads to accumulation of activated Akt in the detergent-insoluble (cytoskeletal) fraction (Supplementary Fig. 8). The silencing of R-Ras reduced activated Akt from this fraction, demonstrating the importance of endogenous R-Ras signaling for the localization of activated Akt. In comparison, activated Akt in the cytoskeletal fraction remained at a basal level upon stimulation by VEGF. These findings are important as they may help us understand how Akt may play multifaceted roles in angiogenesis. While our findings leave new questions about biology of Akt, we believe that these are extremely important questions to raise here for future studies to investigate further. We, therefore, included them in the manuscript.

7. Does constitutive active Akt result in the same phenotype/outcome on microtubules? Again, is R-Ras acting exclusively through Akt or are other players involved downstream R-Ras?

Although our work clearly demonstrated the importance of the R-Ras-Akt axis, other R-Ras pathways are also likely to contribute to endothelial lumenogenesis, for instance, through enhancement of integrin adhesion and VE-cadherin-mediated cell-cell adhesion. We commented on this in the *Discussion* section.

We examined the effect of constitutively active myr-Akt1 on EC lumenogenesis in 3D fibrin gel culture. The expression of myr-Akt1 caused large dome-like structures rising from the EC-coated beads. Thus, it indeed induced lumen formation; however, it failed to make vessel-like tubular structures. It was rather similar to the dome-like lumen formation by MDCK epithelial cells in 2-D culture. The intracellular localization of myr-Akt is probably highly artificial because it is engineered to have Src-like myristoylation. Also, since the overexpression of myr-Akt could activate multiple Akt pathways (both canonical and noncanonical as well as non-physiological) simultaneously, we concluded that this experiment is uninterpretable. Therefore, we did not pursue this strategy for further analyses.

8. The *in vivo* data is nice, but the use of PODXL as a mean to visualize lumenized vessels not as convincing. A combination of intravascular lectins and PECAM should be included to ensure that indeed lumen is compromised. Alternatively, vasculature in skeletal muscle can be very nicely observed *in vivo* by videomicroscopy of fluorescent endothelial cells with fluorescently labeled blood cells (this can be an alternative).

We analyzed vessel lumenogenesis *in vivo* by three different ways. The intravascular lectin/CD31 (PECAM) double staining as suggested (Fig. 4e, 5e, and 6d) as well as EM ultrastructural analysis (Fig. 4b and 5c) in addition to PODXL/CD31 staining (Fig. 4a, 5b, and 6c). All these analyses confirmed the effect of R-Ras on vessel lumenogenesis.

9. As stated previously, it will be important to show the status of microtubules in the KO mice.

We demonstrated this in Fig. 4d (R-Ras KO endothelium staining) and Fig. 4c (western blot analysis of ECs isolated from R-Ras KO vessels in the ischemic GC muscles) as well as Fig. 6b (R-Ras KO vessels rescued by *in vivo* R-Ras transduction).

Reviewer 3

1. It is already known that: i) R-Ras activates PI3K (Marte et al., 1997, *Curr Biol* 7, 63-70); ii) PI3K signals through Akt (Onishi et al., *Genes Cells*, 2007, 12:535-546) and GSK-3 β (Zumbrunn et al., 2001, *Curr. Biol.*, 11:44-49) to stabilize MTs; iii) microtubule assembly and tubulin modifications control EC lumen formation (Kim et al., *Blood*. 2013, 121:3521-3530). Therefore, it is not so innovative to describe a link between the RRas/PI3K/Akt/GSK-3 β /MT stabilization signaling and endothelial lumenogenesis.

Our study reported here addresses three unsolved key questions in vascular biology. First, the molecular mechanism of endothelial lumen formation has not been clearly understood despite its fundamental importance in blood vessel formation. Second, Akt has been widely known as a proangiogenic molecule that strongly induces vessel sprouting and permeability; however, there are several reports that suggest other, sometimes opposing, roles of Akt in angiogenesis. These apparently contradicting activities of Akt raises a question how canonical and non-canonical pathways of Akt may produce divergent effects to coordinate together the formation of functional blood vessels. As we discussed in *Introduction* of the manuscript, the severely impaired ischemic muscle reperfusion of Akt-knockout mice (Ackah et al., 2005) cannot simply be explained by reduced vessel sprouting activity. These previous observations suggest that there are yet unknown functions of Akt in blood vessel formation. Lastly, R-Ras suppresses angiogenic sprouting/branching and stabilizes the vessel wall integrity to promote maturation of regenerating blood vessels (Sawada et al., 2012). However, R-Ras also activates Akt, a strong angiogenesis/vascular permeability-inducer. Hence, there is an apparently contradicting relationship between Akt and R-Ras, and investigating this relationship would be crucial for better understanding of the biology of angiogenesis. We explored these questions in this report. We show

that Akt activation by R-Ras, but not the activation by VEGF, promotes endothelial lumenogenesis, and we demonstrate the significance of this pathway for ischemic muscle reperfusion.

Previous studies investigated the pathway in pieces in different cell types in different biological contexts, all of which are *in vitro*. However, no study has ever connected these pieces together and identified its role in the physiologically essential process of lumenogenesis. Moreover, to our knowledge, a direct role of Akt has never been demonstrated in endothelial or epithelial lumenogenesis. With all these findings, we believe that the message conveyed in this report is novel and highly significant.

2. In figures 1 and 2, the analysis of EC morphogenesis in vitro is not careful and no robust conclusion can be drawn about the possible role of R-Ras in the lumenization of endothelial sprouts. Phase contrast microscopy coupled to fluorescence microscopy, in the absence of a detailed confocal analysis along EC Z-axis of well known apical domain markers, such as podocalyxin (PODXL; Horvat et al., J. Cell Biol., 1986, 102:484-491), do not formally demonstrate that structures lined by ECs in 3D fibrin are hollow tubes. As far as they appear, the hypothetical lumens may for example just be bent chains formed by ECs. Electron microscopy coupled to PODXL immunogold labeling (Horvat et al., J. Cell Biol., 1986, 102:484-491) would be extremely helpful here.

In the revised manuscript, we added confocal 3-D images of lumenized EC sprouts (Supplementary Fig. 1a). In these images, Z-sections were taken up to the half way of the sprout's diameter so that we can observe the interior of cylindrical structure of the EC sprout. The Supplementary Video 1 and 2 have rotating images to show the vessel wall (view from outside) and lumen (view of interior). We also included confocal images of PODXL staining (Supplementary Fig. 1c). In addition, we provide EM pictures of lumenized blood vessels for *in vivo* analyses (Fig.4b, 5c).

3. Authors write “Many of these sprouts showed that they have begun forming lumens (Figure. 1). However, these lumens are disconnected, and the sprouts are incompletely lumenized (Figure. 1). The expression of constitutively active R-Ras (R-Ras38V) dramatically enhanced the EC lumenogenesis to create well-defined, uninterrupted lumens (Figures. 1).” Supposing that authors provide a formal and clear demonstration that R-Ras controls lumen formation, afterwards lumenogenesis must be thoroughly quantified in basal conditions as well as after R-Ras38V overexpression and endogenous R-Ras knock-down. Furthermore, the ability of exogenously-delivered wild type or mutant (e.g. Akt-independent D64A) R-Ras constructs to rescue the disruption of lumenogenesis caused by R-Ras silencing should be verified.

We added quantitative comparisons of lumen formation (Supplementary Fig. 1b) in the revised manuscript. For the second question, we conducted such studies *in vivo* by rescuing the disruption of lumenogenesis caused by R-Ras knockout by *in vivo* transduction of R-Ras (Fig. 5, 6).

4. Authors write “The microtubule stabilizing effect of R-Ras was demonstrated by markedly increased acetylation and deetyrosination of α -tubulin as well as pronounced elongation of microtubules that reach the cell membrane. Moreover, a considerable accumulation of delta 2-tubulin was evident (Figure. 1) indicating the formation of long-

lasting, highly stable microtubule cytoskeleton network in these cells. The R-Ras-dependent microtubule stabilization was also observed in endothelial sprouts in the 3-D culture that exhibited enhanced lumenogenesis and uninterrupted capillary-like tubular structures (Figure. 1).” All these effects must be quantified and not merely described. Which is the quantitative extent of modification of α -tubulin acetylation and detyrosination? How MT elongation was quantitatively defined? In how many ECs was MT elongation quantified? Statistical analysis must be provided as well.

We included quantitative analyses of microtubule modifications with ANOVA statistics in the revised manuscript (Supplementary Fig. 3b). The ratio of post-translationally modified α -tubulin (acetylated α -tubulin or delta 2-tubulin) to the total α -tubulin was determined from immunofluorescence staining of the cells. For this purpose, the positive area for each immunostaining was determined in each cell by Volocity® software. Furthermore, western blot analyses also provide quantitative comparisons of α -tubulin acetylation between the groups (Fig. 1c, 2b, c, e, g). We did not determine each microtubule length; however, based on the immunostaining of α -tubulin or end-binding protein EB1, it is evident that the microtubules extend and reach all the way to the membrane periphery upon upregulation of R-Ras signaling, and they are shorten by R-Ras silencing. We observe these changes in the majority of cells in the culture. In the manuscript, we present several representative images of these cells in multiple figures (Fig. 1d, e, f, Supplementary Fig. 3a, 4). We added scale bars for comparison.

5. Authors write “In the lumenized sprouts, microtubules elongated from the microtubule organizing center positioned at the apical side of the nucleus in each cell, indicating the polarization of ECs (Figure. 1).” Neither an apical domain nor a microtubule organizing center marker was stained to support authors’ statements. Furthermore, it appears that microtubules belonging to different opposite ECs get in contact: this would suggest that, differently from what hypothesized by authors, endothelial tubes of R-Ras38Voverexpressing ECs do not contain any lumen, but their structure is just due to a different 3D organization of ECs and their MTs, compared to tubes formed by control ECs.

We included γ -tubulin staining to indicate the position of microtubule organizing center relative to the nucleus (Supplementary Fig. 5). We provided confocal 3D image and video to demonstrate the presence of lumen (Supplementary Fig. 1a, Supplementary Video 1, 2).

6. In Figure 3, it is not correct to compare the activation of the pAkt/pGSK-3 β /Ac- α -tubulin pathway caused by chronic R-Ras38V overexpression and acute VEGF-A stimulation. The correct experiment would be to compare pAkt/pGSK-3 β /Ac- α -tubulin activation in control and R-Ras silenced ECs that were stimulated or not with VEGF-A. In such a way, authors would understand if R-Ras is involved or not in the transient pAkt/pGSK-3 β /Ac- α -tubulin activation that is elicited by acute VEGF-A stimulation.

We treated control or R-Ras-silenced ECs with or without 50 ng/ml VEGF-A for 30 min. This study showed that VEGF-A can transiently activate Akt, phosphorylate GSK-3 β , and increase acetylation of α -tubulin independently of R-Ras, further supporting the idea that the VEGF-Akt and R-Ras-Akt axes are two separate pathways. We included this result in the revised manuscript (Fig. 3c).

*Note. The basal level of pAkt (VEGF-) is significantly lower in R-Ras-silenced cells than in control cells, which causes a lower pAkt level in VEGF+ R-Ras-silenced cells than in VEGF+ control cells.

7. Authors write “The immunofluorescence staining of ECs indicated that the Ser 473-phosphorylated Akt localizes at the perinuclear late endosomes upon cell stimulation with VEGF-A (Figure. 3)”. No late endosome marker is provided to support this statement.

We revised this sentence to “perinuclear region”. The perinuclear localization pattern of activated Akt has been demonstrated previously (Andjelkovic et al, 1997; Adini et al., 2003; Wheeler et al., 2015).

8. Speaking about revascularization of ischemic skeletal muscles in R-Ras KO mice, authors write “However, the immunoreactivity for podocalyxin (PODXL), which highlights endothelial lumen, was frequently absent in these vessels indicating a deficiency in lumenogenesis (Figure. 4).” Authors must take into account that the acquisition of EC apicobasal polarity usually relies on the delivery of PODXL-containing vesicles to the future apical plasma membrane, rather on the modulation of PODXL expression (for review see Zeeb et al., Curr. Opin. Cell Biol., 2010, 22:626–632). In ECs, is R-Ras controlling PODXL protein levels rather apical delivery from endosomal stores? If yes, is R-Ras acting on PODXL protein at transcriptional, translation, or degradative level? These aspects may also be easily addressed in cultured ECs. PODXL immunogold labeling electron microscopy on muscle sections of R-Ras wild type and KO mice would be also extremely helpful.

This is exactly the reason why we used the term “immunoreactivity” instead of “expression” of PODXL in the original manuscript. To avoid confusion, we revised the sentence as follows: “However, the pattern of immunostaining for podocalyxin (PODXL), which highlights endothelial lumen, was frequently disrupted in these vessels indicating a deficiency in lumenogenesis (Fig. 4a).”

In immunofluorescence of tissue sections, the detection sensitivity often depends on clustering or accumulation of the target molecule in a certain structure such as cell surface, depending on the target molecule or antibody used. The PODXL tissue staining is widely used to indicate the presence of luminal surface in which PODXL accumulates. We used this method. We confirmed by western blot analysis of PODXL that there is no change in the total PODXL expression level by R-Ras transduction or knockdown in ECs (data not shown). Furthermore, we provide EM ultrastructure analyses to demonstrate abnormal lumen-less vessels in R-Ras KO mice (Fig. 4b) and its rescue by R-Ras *in vivo* transduction (Fig. 5c). These observations are also accompanied by the analyses of lectin perfusion, which also indicates the presence of lumen (Fig. 4e, 5e). Thus, multiple studies indicate the role of R-Ras in endothelial lumenization.

9. Most importantly, since R-Ras is an effective activator of anti-apoptotic PI3K-Akt signaling, the impaired blood vessel re-perfusion observed in the muscle of R-Ras KO mice may be due EC apoptosis. Author must hence disprove the fact that lack of R-Ras may significantly increase EC apoptosis elicited by stressful events, such as femoral artery

ligation. EC survival rather than blood vessel lumenogenesis may be the mechanism by which R-Ras influences post-natal angiogenesis.

As discussed above, we demonstrated repeatedly and by multiple ways the importance of R-Ras for EC lumenogenesis of regenerating blood vessels. Obviously, EC lumenogenesis is a prerequisite for blood flow in the new vessels, and the disruption of this process results in impaired tissue reperfusion. Indeed, we demonstrated that R-Ras-dependent vessel lumenization is accompanied by increased blood perfusion of new vessels (Fig. 4e, 5e, 6d). The importance of EC survival and lumenogenesis are not mutually exclusive: both contribute to the functionality of new vessels. However, we found that R-Ras KO mice exhibit significantly increased overall vascularity in the ischemic muscles (Supplementary Fig. 10a-c). We also observed a >2 fold increase of BrdU-positive proliferating ECs in these mice compared with wild type control mice (Supplementary Fig. 10d). These observations suggest that the potential influence of R-Ras on EC survival does not play a major role in the deficient tissue reperfusion observed in the R-Ras KO mice.

10. Scale bars are lacking in Figure panels 1d, 1e, 1f, 1i, 2d, 2h, 2f, 2i, 3a, 3c, 6b, S1a, S2a, S2b, S3b, and S9b.

We added scale bars to the figures where they are lacking.

11. “The cDNA for an R-Ras mutant (R-Ras38V D64A), which is constitutively active but incapable of activating PI3K (ref.3)”. The correct reference is not ref. 3, but ref. 33.

The Methods section has a separate list of references. Ref. 3 is correct.

Reviewers' comments:

Reviewer #1 (Remarks to the Author):

The authors have adequately addressed my comments on the original manuscript.

Reviewer #2 (Remarks to the Author):

Overall the authors have addressed many of the concerns raised by the peer-review process. While the basic bits of information have been known (as discussed by reviewer 3), the essential contribution of R-Ras in the regulation of microtubule stabilization for lumen formation has not been previously revealed. The fact that the authors presented *in vivo* evidence to support this fundamental observation is, in my mind, critical and sufficient. Importantly, the question as to how Ras (vs VEGF) activates Akt to alter microtubule stability remains open and was not answered by the study. Regardless, after evaluation of the data presented and the previous arguments made in the review process, the bulk of information is important, convincing, and propels the field forward.

One important suggestion: the authors should provide quantification of the data presented in Figure 1. Only with robust quantification would the readers have a sense of the biological impact of R-Ras in lumen formation.

Also note that figure 5e (the y axis should read Lectin Perfusion (not Lection perfusion)).

Reviewer #3 (Remarks to the Author):

Authors worked to address many of the open and unexplained issues present in the original version of the manuscript. However, even if the revised version of the manuscript by Li and colleagues is improved, some questions, already present during the first round of review, are still unanswered.

1. The quality of the picture documenting the presence/absence of lumen in cultured ECs is not fully convincing yet. Author should show confocal microscopy pictures acquired along the xz axis of the *in vitro* formed EC sprouts. This will allow evaluating more distinctly the existence of a lumen in control, but not R-Ras silenced ECs? In other words, authors should section the EC sprouts transversally rather than sagittally and a blood vessel should appear as a ring (e.g. see Fig. 4, panel A, 3rd picture from the left of Kim et al., Blood 121:3521-3530).

2. Figure 3c. As authors state in the note to point 6 of their rebuttal letter "The basal level of pAkt (VEGF-) is significantly lower in R-Ras-silenced cells than in control cells, which causes a lower pAkt level in VEGF+ R-Ras-silenced cells than in VEGF+ control cells." However, since this experiment is crucial to support the existence of a VEGF-dependent and a VEGF-independent/R-Ras dependent pathway of Akt activation, data must be carefully quantified. In particular:

A. How was shR-Ras introduced in ECs, plasmid transfection or lentivirus-mediated transduction? Control (Cont) ECs are not transfected or not transduced ECs or are they mock transfected or mock transduced ECs? This aspect should be clarified. If they are not transfected or not transduced ECs, they are not the appropriate control.

B. Western blot bands of phosphorylated or acetylated proteins must be quantified and normalized on their corresponding total protein. This will allow understanding if both in control silenced ECs

and in R-Ras silenced ECs VEGF is eliciting the same proportional increase in pAkt, pGSK-3 β , and acetylated α -tubulin.

C. How many times was this experiment performed? In the reporting check list the corresponding author checked to confirm that "a statement of how many times the experiment shown was replicated in the laboratory"... "is available in all relevant figure legends (or Methods section if too long)". However, this does not seem to apply to Figure 3c as well as to all other Figures describing in vitro data (an exception being Suppl. Fig. 1b, in which individual data points are plotted). Concerning Figure 3c, authors should quantify and plot the different Western blot analyses they performed and provide a graph summarizing these data (see point 2B above), while showing a representative experiment of several as they already did.

3. In point 8 of their rebuttal letter authors state: "We confirmed by western blot analysis of PODXL that there is no change in the total PODXL expression level by R-Ras transduction or knockdown in ECs (data not shown)." Since the fact that modulation of R-Ras amounts does not affect total PODXL expression levels is particularly relevant to try to figure out how R-Ras is affecting the absence of PODXL immunoreactivity in skeletal muscles of R-Ras KO mice, these data must be shown. Furthermore, if R-Ras does not affect PODXL expression, how are authors mechanistically explaining the absence of PODXL immunoreactivity in blood vessels during the revascularization of ischemic skeletal muscles in R-Ras KO mice? May authors formally exclude that ECs of lumenless blood vessels of R-Ras KO mice are more easily undergoing programmed cell death?

4. Supplementary Figure 10d. Representative pictured of BrdU stained ECs of GC muscle vasculature of WT and R-Ras KO mice in response to acute hindlimb ischemia must be shown.

Response to Reviewers' Comments

We thank for the encouragement for this manuscript and recognition of the significance of our findings. The followings are responses to remaining questions and concerns from the last review.

Reviewer 2

1. One important suggestion: the authors should provide quantification of the data presented in Figure 1. Only with robust quantification would the readers have a sense of the biological impact of R-Ras in lumen formation.

The quantitative data of lumen formation and microtubule analyses were originally presented in the Supplementary figures. We moved these data to the main Figures (Fig. 1c and f) for more impactful presentations.

2. Also note that figure 5e (the y axis should read Lectin Perfusion (not Lection perfusion)).

The typo was corrected.

Reviewer 3

1. The quality of the picture documenting the presence/absence of lumen in cultured ECs is not fully convincing yet. Author should show confocal microscopy pictures acquired along the xz axis of the in vitro formed EC sprouts. This will allow evaluating more distinctly the existence of a lumen in control, but not R-Ras silenced ECs? In other words, authors should section the EC sprouts transversally rather than sagittally and a blood vessel should appear as a ring (e.g. see Fig. 4, panel A, 3rd picture from the left of Kim et al., Blood 121:3521-3530).

We now included the views of the XZ and YZ planes next to the XY plane (Supplementary Figure 1b). The rings of the lumens look oval (rather than circle) because the X and Y axes are diagonal to the long axis of the EC sprouts. We also included additional snapshots of the rotating 3-D video images at three different angles (0°, 90°, and 180°). For creating these videos, Z-sections were taken up to the half way of the sprout's diameter in order to view the interior of cylindrical structure of the EC sprout. The 0° angle shows the lumen and the 180° angle shows the endothelial wall from the back view.

2. Figure 3c. As authors state in the note to point 6 of their rebuttal letter “The basal level of pAkt (VEGF-) is significantly lower in R-Ras-silenced cells than in control cells, which causes a lower pAkt level in VEGF+ RRas- silenced cells than in VEGF+ control cells.” However, since this experiment is crucial to support the existence of a VEGF-dependent and a VEGF-independent/R-Ras dependent pathway of Akt activation, data must be carefully quantified. In particular:

A. How was shR-Ras introduced in ECs, plasmid transfection or lentivirus-mediated transduction? Control (Cont) ECs are not transfected or not transduced ECs or are they mock transfected or mock transduced ECs? This aspect should be clarified. If they are not transfected or not transduced ECs, they are not the appropriate control.

It was done by lentivirus transduction. The control ECs were mock-transduced with negative control shRNA described previously (Ref. 2). This information is found in the Methods section – Endothelial cell culture, cDNA constructs, and lentivirus transduction.

“R-Ras knockdown was carried out by lentivirus transduction of shRNA or control shRNA as described previously².”

B. Western blot bands of phosphorylated or acetylated proteins must be quantified and normalized on their corresponding total protein. This will allow understanding if both in control silenced ECs and in R-Ras silenced ECs, VEGF is eliciting the same proportional increase in pAkt, pGSK-3 β , and acetylated α -tubulin.

We included the quantifications of the western blot from three independent experiments in Fig. 3c and 3e.

C. How many times was this experiment performed? In the reporting check list the corresponding author checked to confirm that “a statement of how many times the experiment shown was replicated in the laboratory”...“is available in all relevant figure legends (or Methods section if too long)”. However, this does not seem to apply to Figure 3c as well as to all other Figures describing in vitro data (an exception being Suppl. Fig. 1b, in which individual data points are plotted). Concerning Figure 3c, authors should quantify and plot the different Western blot analyses they performed and provide a graph summarizing these data (see point 2B above), while showing a representative experiment of several as they already did.

In the revised manuscript, we included quantitative analyses of the western blot:

- Fig. 3c and 3e
- Supplementary Fig. 3b and 3e
- Supplementary Fig. 6a-d (for Fig. 2 western blot)
- Supplementary Fig. 8b
- Supplementary Fig. 9b

The quantifications of the western blot are from at least three independent experiments. We mentioned this in the figure legends.

3. In point 8 of their rebuttal letter authors state: “We confirmed by western blot analysis of PODXL that there is no change in the total PODXL expression level by R-Ras transduction or knockdown in ECs (data not shown).” Since the fact that modulation of R-Ras amounts does not affect total PODXL expression levels is particularly relevant to try to figure out how R-Ras is affecting the absence of PODXL immunoreactivity in skeletal

muscles of R-Ras KO mice, these data must be shown. Furthermore, if R-Ras does not affect PODXL expression, how are authors mechanistically explaining the absence of PODXL immunoreactivity in blood vessels during the revascularization of ischemic skeletal muscles in R-Ras KO mice? May authors formally exclude that ECs of lumenless blood vessels of R-Ras KO mice are more easily undergoing programmed cell death?

As this reviewer originally stated in his/her comments of the previous review cycle, “the acquisition of EC apicobasal polarity usually relies on **the delivery of PODXL-containing vesicles to the future apical plasma membrane, rather than the modulation of PODXL expression**”. As we discussed in the previous response, the detection sensitivity of immunofluorescence staining often depends on local accumulation or clustering of the target molecule in a certain cellular compartment, such as apical membrane, for the signals to be readily detectable by fluorescence microscopy. This is indeed often the case for staining of tissue sections. Thus, a decrease of PODXL staining can result from reduced PODXL localization and accumulation at apical surface (due to lack of lumens, i.e. lack of apical surface), and it is not necessarily caused by decreased total PODXL expression level in the cells. Thus, this is indeed in agreement with this reviewer’s original comment about PODXL delivery, rather than PODXL expression modulation, being responsible for lumen formation.

We used PODXL staining merely as a marker for lumen formation as it has been used previously in a number of published studies. The question of how R-Ras may be mechanistically affecting the PODXL accumulation is beyond the scope of this manuscript. We believe that such a question should be addressed by a separate study. We added references for PODXL staining used as a lumen-formation marker in the appropriate section of the text.

References for the use of PODXL as a lumen marker:

Strilić, Boris, Tomáš Kučera, Jan Eglinger, Michael R. Hughes, Kelly M. McNagny, Sachiko Tsukita, Elisabetta Dejana, Napoleone Ferrara, and Eckhard Lammert. "The molecular basis of vascular lumen formation in the developing mouse aorta." *Developmental cell* 17, no. 4 (2009): 505-515.

Pelton, John C., et al. "Multiple endothelial cells constitute the tip of developing blood vessels and polarize to promote lumen formation." *Development* 141.21 (2014): 4121-4126.

Hayashi, Makoto, Arindam Majumdar, Xiujuan Li, Jeremy Adler, Zuyue Sun, Simona Vertuani, Carina Hellberg et al. "VE-PTP regulates VEGFR2 activity in stalk cells to establish endothelial cell polarity and lumen formation." *Nature communications* 4 (2013): 1672.

4. Supplementary Figure 11d. Representative pictured of BrdU stained ECs of GC muscle vasculature of WT and R-Ras KO mice in response to acute hindlimb ischemia must be shown.

Representative pictures are shown in the revised Fig. 11d